# LaMoGen: Language to Motion Generation Through LLM-Guided Symbolic Inference

## Abstract

Human motion is highly expressive and naturally aligned with language, yet prevailing methods relying heavily on joint text-motion embeddings struggle to synthesize temporally accurate, detailed motions and often lack explainability. To address these limitations, we introduce LabanLite, a motion representation grounded in the Labanotation system. Unlike black-box text–motion embeddings, LabanLite encodes each atomic body-part action (e.g., a single left-foot step) as a discrete Laban symbol paired with a textual template. This abstraction decomposes complex motions into interpretable symbol sequences and body-part instructions, establishing a symbolic link between high-level language and low-level motion trajectories. Building on LabanLite, we present LaMoGen, a Text-to-**La**banLite-to-**Mo**tion **Gen**eration framework that enables large language models (LLMs) to compose motion sequences through symbolic reasoning. The LLM interprets motion patterns, relates them to textual descriptions, and recombines symbols into executable plans, producing motions that are both interpretable and linguistically grounded. To support rigorous evaluation, we introduce a Labanotation-based benchmark with structured description–motion pairs and three metrics that jointly measure text–motion alignment across symbolic, temporal, and harmony dimensions. Experiments demonstrate that LaMoGen establishes a new baseline for both interpretability and controllability, outperforming prior methods on our benchmark and public datasets. These results highlight the advantages of symbolic reasoning and agent-based design for language-driven motion synthesis.

## 1 Introduction

Human motions convey rich semantics that often correspond to intentions and instructions expressed in natural language. Establishing a precise mapping between language and motion is therefore essential for computational understanding and modelling of human behaviour. Recent approaches (Guo et al., 2022; Tevet et al., 2023; Zhang et al., 2023a) have made promising progress in generating motion from textual descriptions. However, purely text-driven motion generation remains challenging. Existing methods typically operate by aligning text and motion embeddings within a joint latent space, but this space often fails to capture fine-grained semantic relationships, resulting in motions that do not faithfully reflect the input text instructions. The problem becomes more pronounced when the input text differs from the training data, leading to irrelevant or inaccurate motion outputs due to out-of-distribution issues.

Recent works (Huang et al., 2024; Li et al., 2024b) attempt to address this by decomposing the description into multiple tokens, each describing different body parts or sub-action chronology. Nevertheless, these approaches still do not resolve a key issue: when users provide complex, multi-step instructions—such as "walk forward in 5 steps and then walk backward in 3 steps"—the methods tend to encode the entire instruction as one or more text embeddings, which cannot accurately represent the number of steps or capture the explicit temporal order and causal relationships between sub-actions. Several methods introduce explicit signals to guide motion generation, such as using key joint trajectories (Karunratanakul et al., 2023; Wan et al., 2024) or drawing key poses (Liu et al., 2023; Wang et al., 2025). However, these approaches require manual input and specialised user interfaces, limiting their intuitiveness and ease of use.

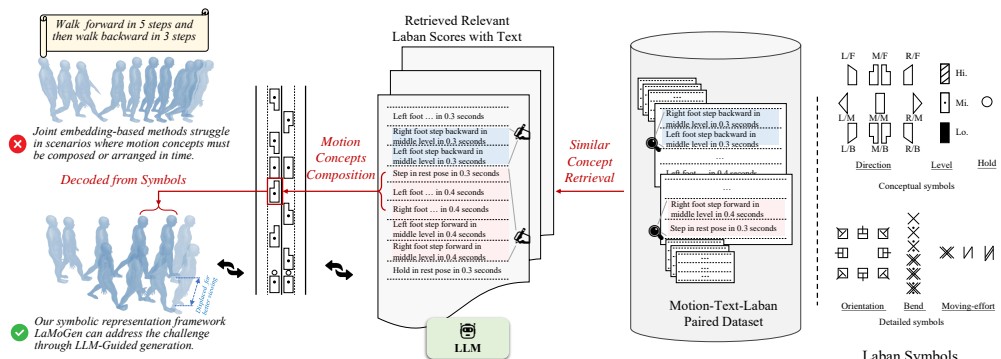

Figure 1: Given a structured text description, methods based on text-motion joint embeddings often fail to generate semantically consistent motion. In contrast, our approach leverages symbolic motion representations, allowing for accurate motion generation. As each symbol is associated with one textual instruction, this design enables LLMs to compose symbolic motion via retrieval augmentation prompting. The right panel illustrates the meaning of each LabanLite symbol.

In contrast, Labanotation (Topaz et al., 1996) provides a motion analysis system that encodes detailed aspects of movement—including which body part is moving, the direction, level, duration, and other qualitative features—into interpretable symbols. By abstracting complex motion into concise symbolic representations, Labanotation enables trained performers to accurately reconstruct intricate motions. For example, the previously mentioned action "walk forward in 5 steps and then walk backward in 3 steps" can be efficiently decomposed into a sequence of symbols representing each atomic movement, as shown in Fig. 1. The combination of these symbols explicitly outlines the temporal and structural characteristics of the entire action. Moreover, if each symbol is associated with a fixed textual description, the motion sequence can be unambiguously translated into a body-part-level instructional description. Therefore, this symbolic system serves as an ideal interface bridging language and motion, clearly delineating both the temporal and spatial structure of actions.

Motivated by these advantages, we propose a novel motion representation *LabanLite*, that faithfully adheres to Labanotation theory, encoding the transition between action states in an interpretable and abstract manner. This unique level of abstraction not only makes the representation human-readable, but also enables direct collaboration with large language models (LLMs) for motion generation. That is, it allows LLMs to actively plan and compose motion sequences by arranging Laban symbols in a retrieval-augmented one-shot, interpretable manner, rather than passively mapping text to motion through black-box embeddings.

Meanwhile, KP (Liu et al., 2024) highlights that commonly used evaluation metrics (Guo et al., 2022) are increasingly indistinct, as some methods even achieve results surpassing those of the ground truth. This observation suggests that current metrics are insufficient for capturing semantic consistency. Building on KP's insights, we introduce a new benchmark based on Labanotation, featuring three metrics that explicitly assess alignment between motion and text, across symbolic, temporal, and harmony dimensions. In contrast to KP's benchmark, which is limited to one or two body parts, our approach assesses all four major body-part groups. This comprehensive evaluation framework promotes greater rigor and transparency within the field.

Building on these foundations, we propose a Text-to-**La**banotation-to-**Mo**tion **Gen**eration framework, dubbed *LaMoGen*. As shown in Fig. 1, to the best of our knowledge, LaMoGen is the pioneering framework that enables LLMs to autonomously compose motion via interpretable symbolic representations. We evaluate LaMoGen on our new benchmark and two standard text-to-motion datasets. Experimental results demonstrate that our framework achieves state-of-the-art performance in aligning textual descriptions with generated motions, highlighting the effectiveness of Labanotation-based representations for both interpretability and controllability in motion synthesis. The main contributions of this work can be summarised as follows. *All code and data are available at https://github.com/xxx/xxx.*

- We introduce LabanLite, a motion representation grounded in Labanotation that abstracts motion into symbolic codes. This enables both explicit text association and faithful trajectory reconstruction.

- We introduce a new paradigm where LLMs act as autonomous agents, actively planning and composing motion sequences via symbolic reasoning. This allows them to address challenges of chronology, compositionality, and precise control over motion attributes that existing frameworks struggle with.
- We propose LaMoGen, a two-level framework that unifies LLM-driven symbolic planning with motion generation over a learnable latent Laban codebook for detail augmentation. LaMoGen delivers superior interpretability and controllability compared to prior approaches.
- We present a comprehensive Labanotation-based benchmark with new metrics and a structured dataset for evaluating alignment between language and motion.

## 2 RELATED WORKS

**Text-based Human Motion Generation** aims to generate diverse, human-like motion from natural language descriptions or action labels, typically by mapping paired textual and motion data into a joint embedding space (Guo et al., 2022; Tevet et al., 2023; Lee et al., 2023; Zhang et al., 2024a; Li et al., 2024b). Subsequent research has focused on abstracting motion signals to extract their semantic content, for achieving a semantically consistent representation between motion and text. Kinematic Phrase (KP) (Liu et al., 2024) proposed a heuristic approach to abstract motions by computing the relative distance between inter-body parts, and used a variational autoencoder to generate motions. However, such methods are limited to low-level signal abstraction and have difficulty capturing high-level semantic meaning. On the other hand, Vector Quantised Variational Autoencoders (Zhang et al., 2023a; Jiang et al., 2024a; Li et al., 2024b; Zeng et al., 2025) have been widely adopted to represent motion as discretised tokens, which can be effectively combined with autoregressive transformers to produce coherent motion sequences. CoMo (Huang et al., 2024) further advances this line of work by discretising motion into pose codes using Posescript (Delmas et al., 2024). However, pose codes represent only the key pose state of individual frames, thus lacking the ability to capture transitional dynamics. To address this limitation, we draw inspiration from CoMo and propose the use of Laban symbols as the intermediate representation between text and motion. Unlike pose codes, each Laban symbol encapsulates not only the starting and ending poses but also the transformation process between them. This makes Laban symbols more abstract and semantically expressive. Consequently, our proposed LaMoGen framework enables LLMs to compose Laban symbols directly, as they are more closely aligned with textual representations and facilitate symbolic reasoning about motion.

**Motion Generation with LLMs**. To our knowledge, no existing work explicitly enables LLMs to autonomously generate motion without fine-tuning (Kalakonda et al., 2023; Zhang et al., 2024b; Zhou et al., 2024). The closest prior works (Shi et al., 2023; Athanasiou et al., 2023; Huang et al., 2024) use LLMs to decompose text into body part descriptions, without allowing them to compose or generate motions. In contrast, we adopt a retrieval-augmented prompting strategy, providing LLMs with reference Laban scores to help them understand how to produce the desired motions.

**Labanotation**. Recent studies primarily use Labanotation as a motion representation for reconstruction tasks. Jiang et al. (2024b) used Laban symbols to explicitly represent and reconstruct inbetweening motions. Li et al. (2024a) mapped hand images to Laban symbols and then to hand motions, to improve hand pose estimation accuracy. In this work, we apply Labanotation to motion generation, bridging textual descriptions and motions.

## 3 METHOD

In this section, we first present LabanLite, an interpretable motion representation derived from Labanotation that encodes detailed body-part movements into symbolic form[1]. Building on this, we introduce LaMoGen, a unified framework that leverages LabanLite to support LLM-driven motion generation. As illustrated in Fig. 2, LaMoGen integrates two core components: a **Laban-Motion**

---

[1]Here, *symbolic* refers to a discretised encoding system aligned with the symbols of an annotation language, each carrying specific semantic and domain-level attributes. This differs fundamentally from continuous-space modelling, which is grounded in raw vector embeddings rather than interpretable symbolic units.

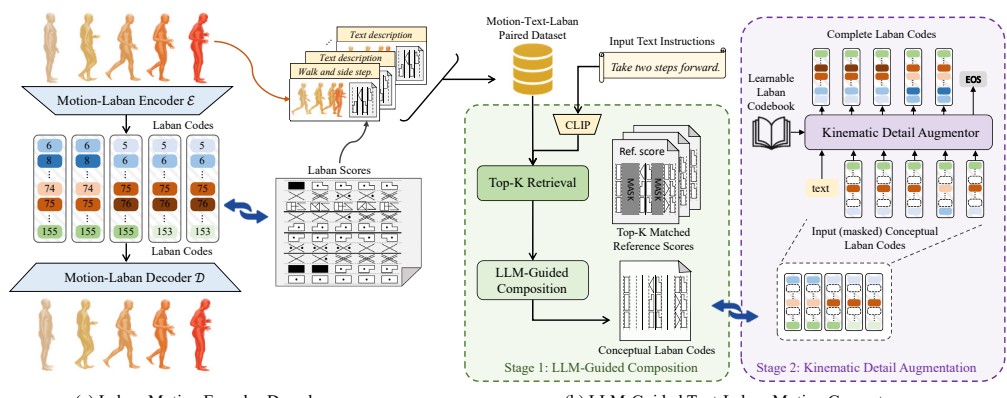

(a) Laban-Motion Encoder-Decoder       (b) LLM-Guided Text-Laban-Motion Generator

Figure 2: Overview of LaMoGen: (a) The Laban-motion Encoder-Decoder enables bidirectional conversion between motion and LabanLite symbols. These symbols are human-readable, as each corresponds to one instructional description, stored in a Text-Laban Database. (b) LLMs perform high-level symbolic planning through retrieval-augmented prompting. The Kinematic Detail Augmentor enriches these plans into temporally coherent motion through autoregressive generation.

**Encoder–Decoder Module**, which enables bidirectional conversion between raw motion data and LabanLite symbols, and a **LLM-Guided Text-Laban-Motion Generation Module**, where LLMs autonomously compose high-level symbolic motion sequences through retrieval-augmented prompting. These symbolic plans are then expanded by a Kinematic Detail Augmentor that enriches them with fine-grained symbols and attributes, ensuring realistic and text-aligned motion generation.

## 3.1 CORE CONCEPTS

### 3.1.1 LABANOTETAION AND LABANLITE

Labanotation is a symbolic system for recording movement, where Laban symbols are arranged on a vertical staff to form a score that captures spatial positions and temporal sequences of body parts (Topaz et al., 1996). A standard staff has eleven columns, each representing a body part, with symbols stacked bottom to top to indicate order and duration. Symbol attributes encode motion details: direction (forward, left, right, etc.) is conveyed by shape, level (high, middle, low) by shading, and additional aspects such as orientation or effort by further markings. Together, these conventions provide a structured and interpretable record of whole-body motion.

Based on Labanotation, we introduce LabanLite, a curated subset designed to preserve expressive richness while being more suitable for digital encoding and motion modeling. As shown in Fig. 1, LabanLite symbols are organized into conceptual and detail categories. Conceptual symbols capture overall motion structures, e.g., *Direction* and *Level*, while *Hold* denotes stationary states. Detail symbols refine these structures with attributes such as body *Orientation*, *Bend*, and *Moving-effort*[2].

LabanLite differs from conventional Labanotation both in symbol selection and data representation. In practice, it focuses on a subset of body-part groups: Left/Right supports, Left/Right leg gestures, Right/Left arms, Right/Left hands, the lower torso (Body), and the Head. Rather than vertical staff columns, each body-part group is represented as a compact record, with attributes encoded directly into fields derived from symbol combinations. This design yields a frame-wise annotation that is both human-interpretable and computation-ready.

**Remark**. One unique strength of LabanLite lies in its temporal expressiveness. Unlike frame-based pose representations (e.g., CoMo (Huang et al., 2024)), LabanLite encodes both intention and duration, allowing LaMoGen to operate over symbolic action units. This design enables the model to interpret and generate compositional instructions such as "*move hand forward in one second*", where symbolic attributes like direction and timing are explicitly encoded and aligned.

---

[2]For example, "wave right hand". Right hand's Direction attribute transitions M/M → R/F → M/M, while Level transitions Lo. → Hi. → Lo., capturing the concept of hand waving. Meanwhile, Right arm's Bend changes from extended to flexed and back, specifying the detailed gesture.

### 3.1.2 LABAN CODE & LABAN CODEBOOK

Labanotation symbols are inherently multi-dimensional: their interpretation depends on the staff location (indicating the body-part group), the vertical axis (encoding time and action duration), and additional attributes (e.g., direction, level, or effort). While this layout is intuitive for human, it complicates digital encoding. To address this, we define each unique symbol assigned to a specific body-part group as a distinct symbol instance and map it to a unique identifier, termed a **Laban code**. Consequently, one Laban symbol can have multiple Laban codes across different body parts.

We collect all $N = 158$ codes into a **Laban codebook** (detail explained in Appendix A.1), denoted $C = \{c_n\}_{n=1}^N$, where each entry $c_n \in \mathbb{R}^{d_c}$ is a learnable embedding. The codebook defines a discrete latent space where each entry is a learned embedding optimised to capture fundamental motion patterns. By combining these embeddings linearly, the system can approximate continuous variations, enabling the composition of complex motions from simpler symbolic building blocks.

### 3.2 LABAN-MOTION ENCODER-DECODER

Given a $T$-frame input motion: $X = \{x_t\}_{t=1}^T$, we propose an Automatic Symbol Detection Workflow $\mathcal{F}$, which converts $X$ into a symbol instance sequence $S = \mathcal{F}(X)$, and $S = \{s_t^{i,j} \mid t \in [1, T], i \in [1, A_j] \ j \in [1, G]\}$. Here, $G$ is the total body-part group number, and $A_j$ is the attribute field dimension[3] for Group $j$. The sequence $S$ is then encoded into an latent vector $Z$ using the Laban codebook $C$. The process will be explained in this section.

### 3.2.1 AUTOMATIC LABAN CODE DETECTION WORKFLOW

The workflow processes each body-part group independently in three steps (see Appendix A.2 for better details). On step one, *Dynamic Interval Segmentation*, we divide motion into coherent time intervals by classifying each frame as either dynamic or stationary (hold) according to the velocity of the end-effector. Frames exceeding a velocity threshold are labelled dynamic, while the rest are considered stationary. This step ensures that symbolic units align with natural atomic actions. On step two, *Frame-wise Symbol Extraction*, we translate motion signals into symbolic attributes. *Direction* and *Level* are measured by the 3D displacement of end-effectors from the pelvis, which are mapped to symbolic categories. *Orientation* is obtained by calculating the Euler angles of the hip vector relative to the negative $y$-axis in both the $xy$- and $yz$-planes and discretising them into eight bins of $45°$ each. *Bend* is captured by discretising Euler angles between adjacent limbs into six bins of $30°$, while *Moving-effort* is quantified by discretising pelvis velocity in the $xy$- and $yz$-planes into five levels. Together, these attributes provide a comprehensive symbolic account of local body dynamics. On step three, in *Interval-wise Symbol Aggregation*, we assign a representative symbol combination to each segmented interval, which appears most frequently within the last $30\%$ of frames in the interval, ensuring that the chosen symbol reflects the stable state of the motion.

### 3.2.2 CODING, DECODING, AND OPTIMISATION OF THE LABAN CODEBOOK

The Laban-Motion Encoder $\mathcal{E}$ transforms the symbol sequence $S = \{s_t^{i,j}\}$, derived from the input motion $X = \{x_t\}$ as described in the previous section, into a latent representation:

$$z_t = \sum_{n=1}^N v_t^n c_n, \tag{1}$$

which is constructed by summing the embeddings of all active codes from the codebook $C = \{c_n\}_{n=1}^N$ at each frame. The binary indicator vector $v_t \in \mathbb{R}^{N \times 1}$ specifies which codes are active: an entry $v_t^n$ is set to 1 whenever the detection workflow $\mathcal{F}$ identifies the corresponding attribute.

We further implement a Laban-Motion Decoder $\mathcal{D}$ that reconstructs motion trajectories from the latent representations $\{z_t\}$, using a standard transformer decoder architecture (Vaswani et al., 2017). The Decoder parameters $\theta$ and the Laban codebook $C = \{c_n\}_{n=1}^N$ are jointly optimized by minimizing the following reconstruction loss $\mathcal{L}_{\text{rec}}$:

$$\{\theta, C\} = \text{argmin}_{\theta, \{c_n\}_{n=1}^N} \mathcal{L}_{rec}(X, \{\mathcal{D}(\sum_{n=1}^N v_t^n c_n; \theta)\}_{t=1}^T), \tag{2}$$

---

[3]Because of varying motion range and function, body-part groups have different attribute field dimensions.

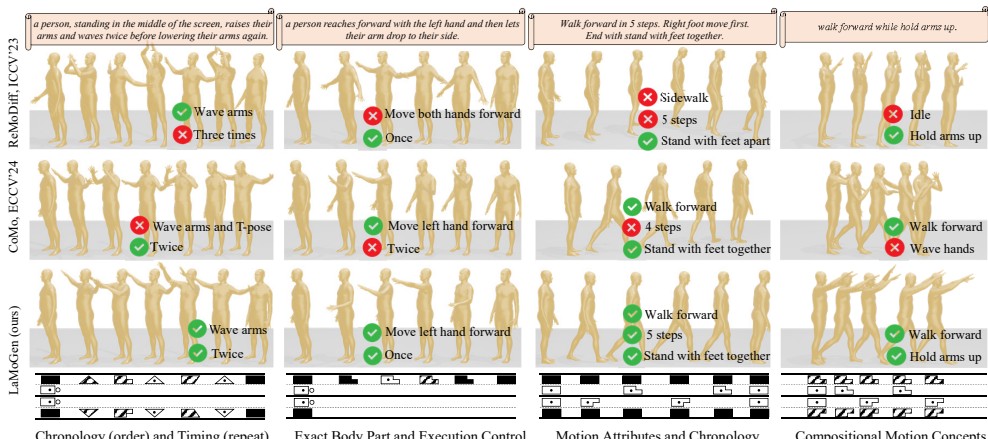

Figure 3: Qualitative comparisons on HumanML3D and Locomotion test sets, with motions progressing from left to right. Misalignments between text and generated motions are highlighted. Note how our method preserves correct sequencing, repetition, and timing of actions, while precisely controlling body parts and motion attributes, and demonstrating compositional generation enabled by LLM-driven symbolic inference—where existing methods fail.

where $\mathcal{L}_{\text{rec}}$ is calculated by summing the $L_1$ distances between poses ($X$ and $\hat{X}$) and velocities ($\dot{X}$ and $\dot{\hat{X}}$) of the input and reconstructed motions: $\mathcal{L}_{rec}(X, \hat{X}) = \|X - \hat{X}\|_1 + \lambda\|\dot{X} - \dot{\hat{X}}\|_1$, with $\lambda$ being a hyperparameter that controls the relative weight of the two terms.

### 3.3 LLM-GUIDED TEXT-LABAN-MOTION GENERATION

Human motion can be understood as the outcome of both conceptual intent and physical execution. In LabanLite, *conceptual symbols* capture high-level motion concepts and intentions (e.g., direction, level, or hold), whose structured patterns can be effectively modeled by large language models. *Detail symbols*, by contrast, specify execution attributes such as orientation, bend, and effort—factors essential for realistic synthesis—which are modeled by a transformer-based architecture (Zhang et al., 2023a) via coding over the learned Laban Codebook. This hierarchical design naturally leads to a two-stage generation pipeline to be introduced in this section.

#### 3.3.1 LLM-GUIDED MOTION CONCEPT COMPOSITION

As introduced earlier, Laban symbols are both visually interpretable and structurally organized, making them particularly well-suited for symbolic reasoning with language models. Their structured form enables LLMs to link textual descriptions with symbolic motion concepts in a manner that is both interpretable and compositional.

Building on this property, the ***first stage*** of our generation framework employs an LLM to compose motion plans at the conceptual level through retrieval-augmented prompting. As shown in Fig. 2(b), we construct a motion database annotated with paired textual descriptions and their corresponding conceptual symbol sequences. At inference time, the user's input text is matched to similar database entries using CLIP-based similarity. The top-$K$ retrieved examples, *presumed to cover the full range of movement elements expressed in the text*, are then supplied to the LLM as in-context examples. By reasoning over the alignment between textual concepts and symbolic motion patterns, the LLM generates a new sequence of conceptual symbols based on the input text.

#### 3.3.2 KINEMATIC DETAIL AUGMENTATION OVER LABAN CODEBOOK

While LLMs are effective at high-level planning, they lack the temporal modelling precision required for detailed motion synthesis. In the ***second stage***, we introduce a Kinematic Detail Augmentor, an autoregressive transformer, to expand the conceptual symbol sequences into *temporally* coherent and *complete* LabanLite codes by adding detail symbols through autoregressive generation.

Table 1: Quantitative comparisons on our Locomotion benchmark, using the proposed Labanotation-based metrics, R-precision Top-3 (R@3) and FID. **Bold** and underlined values indicate the best and the second-best performance, respectively.

| Method | SMT ↑ | | | | TMP ↑ | | | | HMN ↑ | | | R@3 ↑ | FID ↓ |
|---|---|---|---|---|---|---|---|---|---|---|---|---|---|
| | supL | supR | armL | armR | supL | supR | armL | armR | arm-arm | arm-sup | sup-sup | | |
| Real data | - | - | - | - | - | - | - | - | - | - | - | 0.216 | 0.001 |
| MDM (Tevet et al., 2023) | 0.380 | 0.380 | 0.335 | 0.257 | 0.316 | 0.316 | 0.329 | 0.231 | 0.119 | 0.226 | 0.258 | 0.180 | 22.81 |
| ReMoDiff (Zhang et al., 2023b) | 0.470 | 0.470 | 0.427 | 0.395 | 0.377 | 0.377 | 0.385 | 0.322 | 0.179 | 0.264 | 0.351 | 0.192 | 7.121 |
| MoDiff (Zhang et al., 2024a) | 0.491 | 0.491 | 0.470 | 0.411 | 0.362 | 0.362 | 0.412 | 0.328 | 0.180 | 0.281 | 0.361 | 0.196 | 5.701 |
| CoMo (Huang et al., 2024) | 0.358 | 0.358 | 0.474 | 0.382 | 0.211 | 0.211 | 0.284 | 0.250 | 0.203 | 0.298 | 0.252 | 0.176 | 21.94 |
| Ours (Bare) | 0.523 | 0.523 | 0.430 | 0.392 | 0.337 | 0.337 | 0.361 | 0.385 | 0.215 | 0.356 | 0.393 | 0.199 | 5.562 |
| Ours (Qwen3) | 0.571 | 0.571 | 0.478 | 0.495 | 0.401 | 0.401 | 0.456 | 0.448 | 0.369 | 0.334 | 0.450 | **0.212** | 1.903 |
| Ours (DeepSeekV3) | 0.552 | 0.552 | 0.496 | 0.500 | 0.475 | 0.475 | 0.486 | 0.486 | 0.370 | 0.326 | 0.463 | 0.206 | 1.859 |
| Ours (GPT4.1) | 0.583 | 0.583 | 0.493 | 0.476 | 0.507 | 0.507 | 0.501 | 0.492 | 0.303 | 0.367 | 0.508 | 0.208 | 1.861 |
| Ours (Laban) | **0.648** | **0.648** | **0.592** | **0.616** | **0.619** | **0.619** | **0.643** | **0.632** | **0.379** | **0.449** | **0.558** | 0.211 | **1.769** |

We represent the conceptual symbol sequence from the ***first stage*** as masked binary indicator vectors $\hat{v}_{1:t-1}$, where only the predicted conceptual symbol fields are activated. Conditioned on the text input $m$ and $\hat{v}_{1:t-1}$, the Augmentor predicts full binary indicator vectors $v_t$ for each frame, activating embedding entries in the Laban codebook, which encodes both conceptual and detail attributes. The prediction is made over estimated activation probabilities: $p_t^n = P(v_t^n = 1|m, \hat{v}_{1:t-1})$.

During training, conceptual vectors are constructed by masking detailed attributes in the input, with random masking applied to improve generalisation and prevent overfitting. To mark motion termination, an end-of-sequence token <EOS> is appended, extending the codebook to $N + 1$ entries. Through this process, the Augmentor enriches the frame-wise conceptual symbol plans produced by the LLM with detailed attributes, converting them into fully specified Laban codes. The learning objective is defined using a binary cross-entropy loss:

$$\mathcal{L}_{gen} = -\sum_{t,n} \left[ v_t^n \log p_t^n + (1 - v_t^n) \log(1 - p_t^n) \right]. \tag{3}$$

Finally, the Laban-Motion Decoder $\mathcal{D}$ reconstructs the enriched codes into instruction-aligned, detailed motion trajectories.

# 4 LABAN BENCHMARK

To assess the effectiveness of the proposed LaMoGen framework—particularly its ability to capture fine-grained, structured textual instructions and accurately ground them in the generated motion—we introduce a new evaluation benchmark. This benchmark consists of a locomotion–Laban–text paired dataset, together with three Labanotation-based metrics designed to measure symbolic, temporal, and harmonious alignment. A detailed description of the dataset construction and evaluation metrics is provided in Appendix A.3.

**Locomotion–Laban–text Paired Dataset.** We select all locomotion sequences (e.g., walking, running, and jumping) from the AMASS (Mahmood et al., 2019) dataset and decompose them into atomic actions. Annotation of these atomic actions is carried out in a semi-automated manner: locomotion details such as step count, left/right step order, and action labels are manually extracted, after which these details are expanded into natural language descriptions with the assistance of LLMs. LabanLite symbols are then detected using the workflow introduced in Sec. 3.2.1.

**Laban-based Metrics.** Given the professional rigor of Labanotation in annotating and evaluating motion, we propose three complementary metrics, i.e., Semantic Alignment (SMT), Temporal Alignment (TMP), and Harmonious Alignment (HMN), to assess the effectiveness of text-to-motion generation. Unlike the KP benchmark (Liu et al., 2024), which is limited to one or two body parts, our metrics evaluate multi-body consistency across four key body parts (right/left arms and feet). These metrics operate by converting both ground-truth and generated motions into conceptual symbol sequences, followed by intra- and inter-body-part comparisons using the *Longest Common Subsequence* (See mathematical definition in Appendix A.3). Specifically, SMT measures sequence similarity within individual body parts while ignoring duration. TMP extends this by incorporating symbol duration to enforce temporal consistency. HMN further evaluates coordination across multiple body parts by treating co-occurring symbols as joint units. For example, if the ground truth shows the left foot stepping forward while the left arm swings backward, the generated motion is expected to exhibit a comparable synchronised pattern.

Table 2: Quantitative comparisons with state-of-the-art methods on the HumanML3D test and KIT-ML test datasets, under standard protocols. **Bold**, underlined, and *italicised* values denote the best, second-best, and third-best performance, respectively.

| Method | R-precision ↑ | | | FID ↓ | MM-Dist ↓ | Diversity → | Multi-Mod. ↑ |
|---|---|---|---|---|---|---|---|
| | Top-1 | Top-2 | Top-3 | | | | |
| **HumanML3D** | | | | | | | |
| Real data | $0.511^{\pm.003}$ | $0.703^{\pm.003}$ | $0.797^{\pm.002}$ | $0.002^{\pm.000}$ | $2.974^{\pm.008}$ | $9.503^{\pm.085}$ | - |
| Guo et al. (2022) | $0.457^{\pm.002}$ | $0.639^{\pm.003}$ | $0.740^{\pm.003}$ | $1.067^{\pm.002}$ | $3.340^{\pm.008}$ | $9.188^{\pm.002}$ | $2.090^{\pm.083}$ |
| MDM (Tevet et al., 2023) | $0.320^{\pm.005}$ | $0.498^{\pm.004}$ | $0.611^{\pm.007}$ | $0.544^{\pm.044}$ | $5.566^{\pm.027}$ | $\mathbf{9.559^{\pm.086}}$ | $\mathbf{2.799^{\pm.072}}$ |
| ReMoDiff (Zhang et al., 2023b) | $0.510^{\pm.005}$ | $0.698^{\pm.006}$ | *$0.795^{\pm.004}$* | $\mathbf{0.103^{\pm.004}}$ | $\mathbf{2.974^{\pm.016}}$ | $9.018^{\pm.075}$ | $1.795^{\pm.043}$ |
| MoDiff (Zhang et al., 2024a) | $0.491^{\pm.001}$ | $0.681^{\pm.001}$ | $0.782^{\pm.001}$ | $0.630^{\pm.001}$ | $3.113^{\pm.001}$ | *$9.410^{\pm.049}$* | $1.533^{\pm.042}$ |
| CoMo (Huang et al., 2024) | *$0.502^{\pm.002}$* | $0.692^{\pm.007}$ | $0.790^{\pm.002}$ | $0.262^{\pm.004}$ | *$3.032^{\pm.015}$* | $9.936^{\pm.066}$ | $1.013^{\pm.046}$ |
| KP (Liu et al., 2024) | $0.496^{\pm.000}$ | - | - | $0.275^{\pm.000}$ | - | $9.975^{\pm.000}$ | *$2.218^{\pm.000}$* |
| Ours (Bare) | $0.438^{\pm.005}$ | $0.591^{\pm.004}$ | $0.755^{\pm.003}$ | $1.091^{\pm.013}$ | $3.999^{\pm.020}$ | $8.555^{\pm.097}$ | $1.421^{\pm.089}$ |
| Ours (GPT4.1 mini) | $0.453^{\pm.003}$ | $0.679^{\pm.003}$ | $0.779^{\pm.004}$ | $0.561^{\pm.008}$ | $3.717^{\pm.011}$ | $8.434^{\pm.115}$ | $1.263^{\pm.034}$ |
| Ours (GPT4.1) | $0.491^{\pm.002}$ | *$0.694^{\pm.002}$* | $0.796^{\pm.003}$ | *$0.252^{\pm.003}$* | $3.087^{\pm.003}$ | $9.124^{\pm.058}$ | $1.131^{\pm.027}$ |
| Ours (Laban) | $\mathbf{0.513^{\pm.003}}$ | $\mathbf{0.704^{\pm.002}}$ | $\mathbf{0.813^{\pm.006}}$ | $0.206^{\pm.003}$ | $2.993^{\pm.009}$ | $9.635^{\pm.109}$ | $0.973^{\pm.024}$ |
| **KIT-ML** | | | | | | | |
| Real data | $0.424^{\pm.005}$ | $0.649^{\pm.006}$ | $0.779^{\pm.006}$ | $0.031^{\pm.006}$ | $2.788^{\pm.012}$ | $11.08^{\pm.097}$ | - |
| Guo et al. (Guo et al., 2022) | $0.370^{\pm.005}$ | $0.569^{\pm.007}$ | $0.693^{\pm.007}$ | $2.770^{\pm.109}$ | $3.401^{\pm.008}$ | $10.91^{\pm.119}$ | $1.482^{\pm.065}$ |
| MDM (Tevet et al., 2023) | $0.164^{\pm.004}$ | $0.291^{\pm.004}$ | $0.396^{\pm.004}$ | $0.497^{\pm.021}$ | $9.191^{\pm.022}$ | $10.85^{\pm.109}$ | $\mathbf{1.907^{\pm.214}}$ |
| ReMoDiff (Zhang et al., 2023b) | $\mathbf{0.427^{\pm.014}}$ | *$0.641^{\pm.004}$* | $0.765^{\pm.055}$ | $\mathbf{0.155^{\pm.006}}$ | $\mathbf{2.814^{\pm.012}}$ | $10.80^{\pm.105}$ | $1.239^{\pm.028}$ |
| MoDiff (Zhang et al., 2024a) | $0.417^{\pm.004}$ | $0.621^{\pm.004}$ | $0.739^{\pm.004}$ | $1.954^{\pm.062}$ | $2.958^{\pm.005}$ | $11.10^{\pm.143}$ | $0.730^{\pm.013}$ |
| CoMo (Huang et al., 2024) | *$0.422^{\pm.009}$* | $0.638^{\pm.007}$ | *$0.765^{\pm.011}$* | *$0.332^{\pm.045}$* | *$2.873^{\pm.021}$* | $10.95^{\pm.196}$ | $1.249^{\pm.008}$ |
| Ours (Bare) | $0.400^{\pm.004}$ | $0.621^{\pm.004}$ | $0.750^{\pm.003}$ | $0.685^{\pm.005}$ | $3.222^{\pm.011}$ | $11.74^{\pm.140}$ | *$1.305^{\pm.122}$* |
| Ours (GPT4.1 mini) | $0.418^{\pm.005}$ | $0.630^{\pm.004}$ | $0.761^{\pm.006}$ | $0.550^{\pm.005}$ | $3.274^{\pm.011}$ | $11.85^{\pm.175}$ | $1.103^{\pm.051}$ |
| Ours (GPT4.1) | $0.421^{\pm.004}$ | $0.649^{\pm.004}$ | $0.775^{\pm.013}$ | $0.415^{\pm.011}$ | $3.165^{\pm.007}$ | $11.30^{\pm.166}$ | $1.028^{\pm.101}$ |
| Ours (Laban) | $0.424^{\pm.006}$ | $\mathbf{0.657^{\pm.005}}$ | $\mathbf{0.782^{\pm.009}}$ | $0.254^{\pm.004}$ | $2.821^{\pm.094}$ | $\mathbf{11.09^{\pm.184}}$ | $0.979^{\pm.012}$ |

## 5 EXPERIMENTS

### 5.1 IMPLEMENTATION DETAILS

Following standard protocols (Guo et al., 2022), we evaluate our method on HumanML3D (Guo et al., 2022) and KIT-ML (Plappert et al., 2016) datasets. Evaluation metrics include: Fréchet Inception Distance (FID) for distributional similarity (lower is better); R-Precision and Multimodal Distance (MM-Dist) for text-motion correspondence (higher is better); and Diversity and Multi-Modality (Multi-Mod.) for motion variability (Diversity closer to ground truth and higher Multi-Mod. are preferred). For the Laban Benchmark, we evaluate models trained on HumanML3D without fine-tuning, reporting Laban metrics, R-Precision, and FID for comprehensive assessment. All results are averaged over 20 independent runs.

### 5.2 EVALUATION

Tables 1 and 2 present quantitative comparisons between our method and state-of-the-art methods on the Locomotion, HumanML3D, and KIT-ML datasets. We evaluate our model under six configurations: *Bare*, where motion generation is conditioned barely on text, serving as a baseline for assessing text comprehension; *GPT4.1 mini*, *GPT4.1*, *Qwen3* and *DeepSeekV3*, where generation is conditioned by both text and conceptual cues composed by specific LLM versions; and *Laban*, where conceptual cues are derived from ground truth, to simulate human composition.

**Results on Laban Benchmark.** As shown in Table 1, LaMoGen achieves the best performance across both Laban metrics and two standard evaluation metrics, demonstrating superior text-motion alignment compared to other methods. The LLM-assisted configurations show that LLMs can effectively interpret and compose conceptual Laban symbols as explicit guidance, resulting in higher Laban metric scores than other baselines. In contrast, methods such as MDM and CoMo, that rely on joint text–motion embeddings, often fail to align text and motion under lengthy or out-of-distribution inputs, as reflected by their higher FID scores.

**Results on conventional benchmarks.** As shown in Table. 2, LaMoGen consistently ranks top-3 across five standard metrics, highlighting its robust generalisation. We attribute these strong results to the proposed LabanLite representation, which discretises motion into interpretable symbolic sequences. This design enhances the consistency between instructional text and motion, and enables

the model to reason about both inter- and intra-body part relationships. However, although LaMo-Gen excels on most metrics, its FID scores lag behind others. We believe the reason is that its high abstraction uses identical symbols for different low-level variations that share the same high-level semantics. For example, raising the hand from Lo. to Hi. is represented by the same symbols, but different individuals may perform it with varying speeds. Such low-level movement variations are inherently beyond LabanLite's expressive capacity, resulting in higher FID scores.

**Qualitative comparisons.** We evaluate the motion in four aspects: *Chronology*, whether motions occur in the proper sequence, such as which body part moves first; *Timing control*, whether motions follow specified durations; *Explainability*, whether it is clear why motions happen a certain way; and *Compositionality*, whether different body-part actions can be combined. Four examples are shown in Fig. 3, with more comparisons in the Appendix A.5. Our analysis reveals that ReMoDiff and CoMo occasionally violate chronological order and struggle to specify movement timings. For example, CoMo ends in a T-pose in the first example, and ReMoDiff moves the hands three times in the second. Because ReMoDiff and CoMo lack explicit control signals, their motions are less explainable—it is unclear why ReMoDiff moves both hands forward. In contrast, LoMoGen uses symbolic representations that make the movement logic clear, such as when to step forward and which body part performs the step. For compositionality, in the final example, ReMoDiff raises the arms but does not walk, CoMo walks while waving the arms, while our method combines the actions correctly. Across all examples, our method meets all four criteria, showing the advantage of LLM-driven symbolic inference.

## 5.3 ABLATION STUDY

**LLM's capability**. We compare two GPT-4.1 versions to assess the role of LLM intelligence. According to the handbook (OpenAI, 2025), GPT-4.1 is more capable than GPT-4.1 mini. Table 2 shows that higher LLM capability leads to better generation performance, confirming that stronger LLMs are more effective at composing Laban symbols and understanding their relationships.

**Number of top-matched references**. For LLMs, we provide top-matched conceptual examples to guide the composition of new conceptual codes. We investigate how the number of examples affects performance on the HumanML3D test set, using GPT-4.1. As shown in Table 3, increasing the number of examples from 1 to 3 consistently improves performance, indicating that the LLM benefits from having sufficient examples for accurate imitation. But adding more (5 or 7) offers no further improvement, likely due to exceeding the LLM's context window and causing it to forget the most relevant cues. Thus, we use top-3 retrieval as the default setting for optimal results.

Table 3: Ablation study of different numbers of top-matched references and masking ratios on the HumanML3D test set.

| Top ref. | R@3 ↑ | FID ↓ | MM-Dist ↓ | Multi-Mod. ↑ |
|---|---|---|---|---|
| 1 | 0.755 | 0.595 | 3.840 | 0.954 |
| 3 | 0.796 | 0.252 | 3.087 | 1.131 |
| 5 | 0.782 | 0.371 | 3.191 | 1.030 |
| 7 | 0.774 | 0.324 | 3.225 | 0.971 |

| Mask rat. | R@3 ↑ | FID ↓ | MM-Dist ↓ | Multi-Mod. ↑ |
|---|---|---|---|---|
| 0.1 | 0.789 | 0.258 | 3.008 | 1.092 |
| 0.3 | 0.796 | 0.252 | 3.087 | 1.131 |
| 0.5 | 0.787 | 0.299 | 3.211 | 1.011 |
| 0.7 | 0.772 | 0.345 | 3.290 | 0.992 |
| 0.9 | 0.705 | 0.457 | 3.302 | 0.985 |

**Masking ratio on Laban codes**. We also examine how the masking ratio on Laban codes affects performance during code generation. A higher masking ratio reduces the influence of conceptual cues, making motion generation rely more on text and less on Laban symbols, reducing the influence of conceptual guidance. We conduct this study on the HumanML3D test set, with GPT4.1 and top-3 retrieval settings. Comparing results in Table 3, a masking ratio of 0.3 offers the best balance and achieves optimal performance.

## 6 CONCLUSION

In this work, we have introduced LabanLite, a novel, human-interpretable motion representation rooted in Labanotation, and presented **LaMoGen**, a pioneering Text-to-**La**banotation-to-**Mo**tion **Gen**eration framework. By leveraging LabanLite's concise symbolic abstraction, our approach enables large language models to autonomously plan and compose motion sequences through explicit, interpretable symbolic reasoning—moving beyond the limitations of traditional joint embedding methods. The newly proposed Labanotation-based benchmark, together with comprehensive metrics, provides a rigorous, multidimensional evaluation of text-motion alignment in symbolic, temporal, and harmony aspects. Experiments on both our benchmark and public datasets demonstrate that LaMoGen achieves state-of-the-art performance in terms of interpretability and controllability, effectively capturing fine-grained temporal structures and explicit action sequences from natural language instructions.

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

## A APPENDIX

This is the supplementary material, which provides additional details that could not be included in the main paper due to page limitations. Specifically, it covers the following points:

- Detailed description of LabanLite, as described in Sec. A.1, including a more comprehensive definition and the semantic interpretation of Laban symbols.
- Further information on the Automatic Symbol Detection Workflow, as described in Sec. A.2, including the formulated process and the predefined threshold look-up table.
- Further information on the Laban Benchmark, such as the formal definition of the Laban metric computation, the construction of the Locomotion dataset, and illustrative examples, as detailed in Sec. A.3.
- Experiment details, as described in Sec. A.4, including the training details, implementation details of the proposed model, and prompts utilised in large language models.
- Additional experiments and the corresponding discussions, as detailed in Sec. A.5, including a user study that evaluates human preferences among three text-conditioned motion generation results and more supplementary qualitative visual examples, including failure cases.
- Discussion of limitations and future directions, as described in Sec. A.6.

The complete code of our framework and the related data will be released after publication.

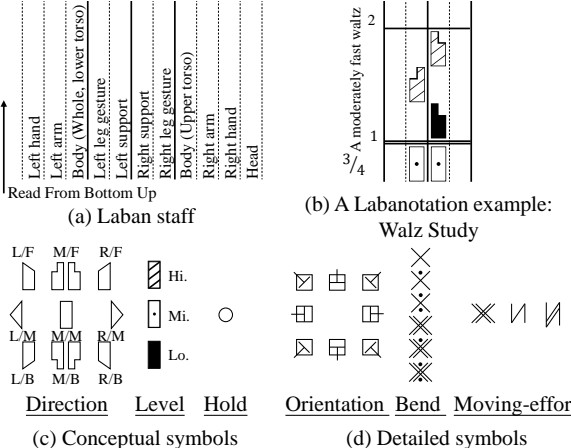

Figure 4: Preliminary of Labanotation. (a) Illustration of a Laban staff. (b) A segment of a Laban score Topaz et al. (1996). (c) Illustration of Conceptual symbols. "L", "R", "F", "M", "B" represent left, right, forward, middle, backward directions. "Hi.", "Mi.", "Lo." represent high, middle, low levels. (d) Illustration of Detailed symbols.

### A.1 DETAILS OF LABANLITE

#### A.1.1 PRELIMINARY

Labanotation employs Laban symbols arranged on a vertical Laban staff to form a Laban score, recording the spatial positions and temporal sequences of different body parts Topaz et al. (1996). As shown in Fig. 4(a), a standard Laban staff has 11 columns, each representing a specific body part. Laban symbols are sequenced in each column to indicate the movement and its duration. Fig. 4(b) shows an example score for a waltz study. Choreographers align the staff with music using bar lines and time signatures.

LabanLite is a curated subset of Labanotation that retains its expressive power while being explicitly designed for digital encoding and motion modelling. Specifically, we refine the Labanotation system according to the following principles, to facilitate both human interpretability and computational readiness.

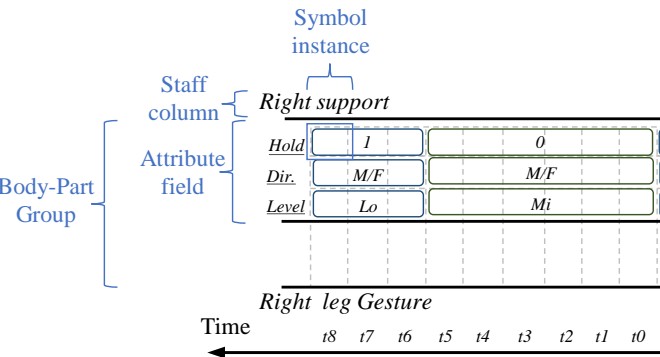

Figure 5: Illustration of the relationships among Body-Part Groups, attribute fields, Laban staff columns, and symbol instances.

**Body-Part Group and Movement Attributes.** We extend the original definition of the Laban staff column from Labanotation by introducing the concept of Body-Part Groups. In LabanLite, each column is restructured to represent a Body-Part Group, and each group is associated with a set of movement attributes. These attributes correspond to specific sets of Laban symbols that are placed within a given column.

Notably, a single Body-Part Group may be represented by multiple staff columns, reflecting the fact that a group can consist of multiple body parts, each with distinct attributes and associated Laban symbols. The reason for this design is that, due to kinematic constraints, not every staff column (i.e., body part) possesses the same set of attribute fields. For example, the Level symbol is not applicable to the Head column, as the head cannot independently move to arbitrary heights; thus, the Head does not have the Level attribute. This design enables the simultaneous recording of both conceptual and detailed movement information. The detailed definitions of Body-Part Groups and their attributes are provided in Table 13.

To be more intuitive, for example, consider the movement "wave left hand." The relevant Body-Part Group is "Upper-L," which includes the staff columns for the "Left arm" and "Left hand." The Direction, Level, and Hold symbols are assigned to the "Left hand" column to indicate the high-level movement concept (e.g., raising up the left arm), while the Bend symbol is placed on the "Left arm" column to capture lower-level movement details (e.g., flexing and stretching the left arm).

**Conceptual and Detailed Laban Symbols.** In LabanLite, we categorise Laban symbols into two types:

- Conceptual symbols, including Direction, Level, and Hold, which describe general movement concepts and structural aspects of motion;
- Detailed symbols, including Orientation, Bend, and Moving-effort, which capture subtle details of individual body part movements.

Tables 7, 8, 9, 10, 11, and 12 show the names, graphical appearances, and *partial semantic meanings* of Laban symbols: Direction, Level, Hold, Orientation, Bend, and Moving-effort, respectively. Please note that, we refer to them as "partial" semantic meanings because the full meaning of a symbol depends on additional factors such as which Body-Part Group it belongs to, its length, and its position on the Laban staff. Please also note that for the M/F and M/B symbols in the Direction category, each typically has two graphical forms. The specific form used depends on the body part indicated by the staff column, e.g., if the M/F symbol is placed in the "Left hand" column, its left-side form is used; if placed in the "Right hand" column, its right-side form is used.

A.1.2 QUICK TUTORIAL OF LABANOTATION

To facilitate readers' understanding of Labanotation, we provide a brief tutorial as a prerequisite for comprehending LabanLite. In this section, we explain how to read a Laban score and introduce its

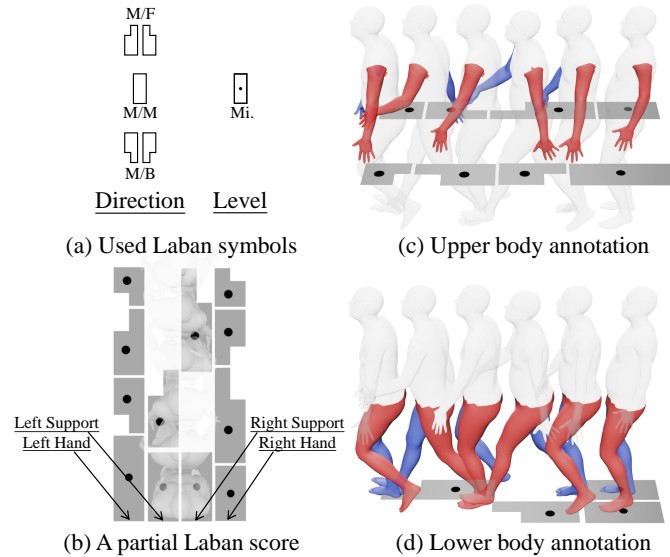

(a) Used Laban symbols

(c) Upper body annotation

(b) A partial Laban score

(d) Lower body annotation

Figure 6: Illustration of a partial Laban score. This figure provides a visual explanation of the annotation process for a forward walk movement.

annotation protocol, thereby helping readers quickly grasp the essentials of Labanotation and then understand the LabanLite.

In Labanotation, certain constraints and simplifications are applied to the symbol sequence to facilitate human readability, resulting in a set of specific visualisation rules:

1. Columns that are not of interest can be left blank, allowing dancers to interpret and perform these unspecified movements at their own discretion. For instance, if a choreographer wishes to emphasise a leap, the lower body columns (Left support and Right support) would contain relevant symbols, while the arm columns may be left empty, granting the dancer creative freedom for upper body movement.

2. For the Left Support and Right Support columns, when a symbol appears in one column at a particular time, the corresponding position in the opposite foot's column is typically left empty, as illustrated in Fig. 6(b).

3. Direction and Level symbols are often merged into a single symbol for simplicity, as illustrated in Fig. 6(b).

4. A new Laban symbol is only added to a column when it differs from the previous symbol; otherwise, the symbol is omitted to maintain clarity and conciseness in the Laban score.

To be more intuitive, Fig. 6 presents a partial Laban score depicting a forward walk. As illustrated, this version of the Laban staff consists of four columns, from left to right representing the Left hand, Left support, Right support, and Right hand. This Laban score abstracts a relatively complex motion into a simple Laban symbol sequence using only four types of symbols: Level Mi., Direction M/F, M/M and M/B. Through this walk forward movement, the initial state features both feet together and hands naturally lowered, corresponding to all four columns remaining in their initial positions: Direction M/M with Level Mi.; the left foot then steps forward accompanied by a backward swing of the right arm to complete the first step, which is denoted by a forward-medium symbol set (Direction M/F and Level Mi.) in the supporting column and a backward symbol set (Direction M/B and Level Mi.) in the hand column. The final state is characterised by the left foot stepping forward (Direction M/F and Level Mi.) while the right foot remains behind.

In the following, we describe the proposed procedure for instantiating Laban symbols, which enables a Laban score to be computational-ready, and the reason behind.

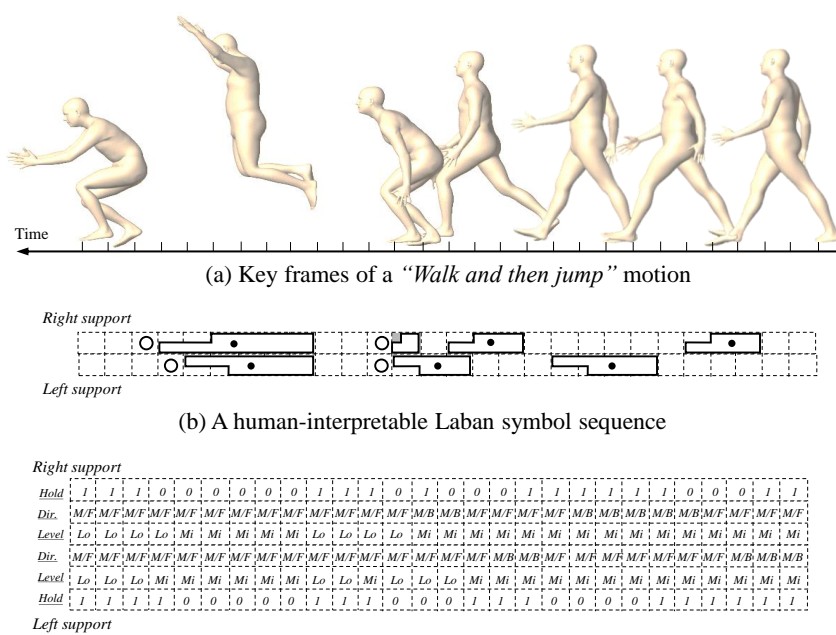

(a) Key frames of a *"Walk and then jump"* motion

*Right support*

*Left support*

(b) A human-interpretable Laban symbol sequence

*Right support*

| Hold | 1 | 1 | 1 | 0 | 0 | 0 | 0 | 0 | 0 | 0 | 1 | 1 | 1 | 0 | 1 | 0 | 0 | 0 | 1 | 1 | 1 | 1 | 1 | 1 | 1 | 0 | 0 | 0 | 1 | 1 |
|---|---|---|---|---|---|---|---|---|---|---|---|---|---|---|---|---|---|---|---|---|---|---|---|---|---|---|---|---|---|---|
| Dir. | M/F | M/F | M/F | M/F | M/F | M/F | M/F | M/F | M/F | M/F | M/F | M/F | M/F | M/F | M/B | M/B | M/F | M/F | M/F | M/B | M/B | M/B | M/B | M/B | M/F | M/F | M/F | M/F | | |
| Level | Lo | Lo | Lo | Lo | Mi | Mi | Mi | Mi | Mi | Mi | Lo | Lo | Lo | Lo | Mi | Mi | Mi | Mi | Mi | Mi | Mi | Mi | Mi | Mi | Mi | Mi | Mi | Mi | | |
| Dir. | M/F | M/F | M/F | M/F | M/F | M/F | M/F | M/F | M/F | M/F | M/F | M/F | M/F | M/F | M/F | M/F | M/B | M/B | M/F | M/F | M/F | M/F | M/F | M/F | M/F | M/B | M/B | M/B | | |
| Level | Lo | Lo | Lo | Mi | Mi | Mi | Mi | Mi | Mi | Lo | Lo | Mi | Lo | Lo | Lo | Mi | Mi | Mi | Mi | Mi | Mi | Mi | Mi | Mi | Mi | Mi | Mi | | | |
| Hold | 1 | 1 | 1 | 1 | 1 | 0 | 0 | 0 | 0 | 0 | 0 | 0 | 1 | 1 | 1 | 1 | 0 | 0 | 0 | 1 | 1 | 1 | 1 | 0 | 0 | 0 | 0 | 1 | 1 | 1 |

*Left support*

(c) A computational-ready Laban symbol instance sequence

Figure 7: Visual comparison between a human-interpretable symbol sequence and a computational-ready symbol instance sequence. The illustrated Laban staff columns correspond to Left support and Right support.

### A.1.3 SYMBOL INSTANTIATION

Laban symbols are used to describe the state of specified body-part actions. In other words, a Laban symbol is only meaningful when its associated Body-Part Group (i.e., the relevant staff column), the position along the staff vertical axis (indicating the action's beginning time), and its duration (specifying how long the action persists) are all specified.

To make the symbol sequence computational-ready, we define a symbol instance as the unique pairing of a symbol and a Body-Part Group, representing a specific action instance in frame-wise annotation. The relationship among Laban symbols, staff columns, symbol instances and Body-Part Groups is illustrated in Fig. 5.

Specifically, as described in the main paper, the instantiation process is formulated as follows. Given an input motion sequence of $T$ frames, $X = \{x_t\}_{t=1}^{T}$, we propose an Automatic Symbol Detection Workflow $\mathcal{F}$, which converts $X$ into a sequence of symbol instances $S = \mathcal{F}(X)$, where:

$$S = \{s_t^{i,j} \mid t \in [1, T], \ i \in [1, A_j] \ j \in [1, G]\}. \tag{4}$$

Here, $G$ denotes the total number of Body-Part Groups, and $A_j$ represents the attribute field dimension for Group $j$. Note that, different Body-Part Groups may be associated with distinct sets of motion attribute fields, due to variations in their movement ranges and functional roles.

**Human-interpretable vs. Computational-ready.** In Labanotation, as elaborated before, specific constraints and simplifications are applied to the symbol sequence to enhance human readability and ensure the Laban score remains clear and concise. In contrast, for computational purposes (e.g., decoding Laban codes into motions, and generating Laban codes), it is necessary to instantiate every Laban symbol so that each attribute field for every Body-Part Group has an explicit value at every frame, without any omissions. As detailed in the main paper, we address this requirement by introducing the concept of *symbol instances*. A comparison between the human-interpretable and computational-ready Laban symbol instance sequences is illustrated in Fig. 7, which depicts the movement of the lower body. This figure highlights the differences between a human-readable Laban score and the proposed computational-ready Laban symbol instance sequence.

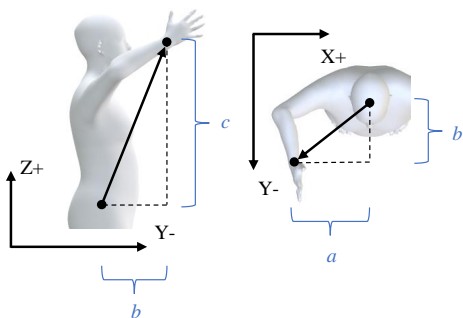

Figure 8: Illustration of extracting Direction and Level symbols for the right hand, in a canonical space.

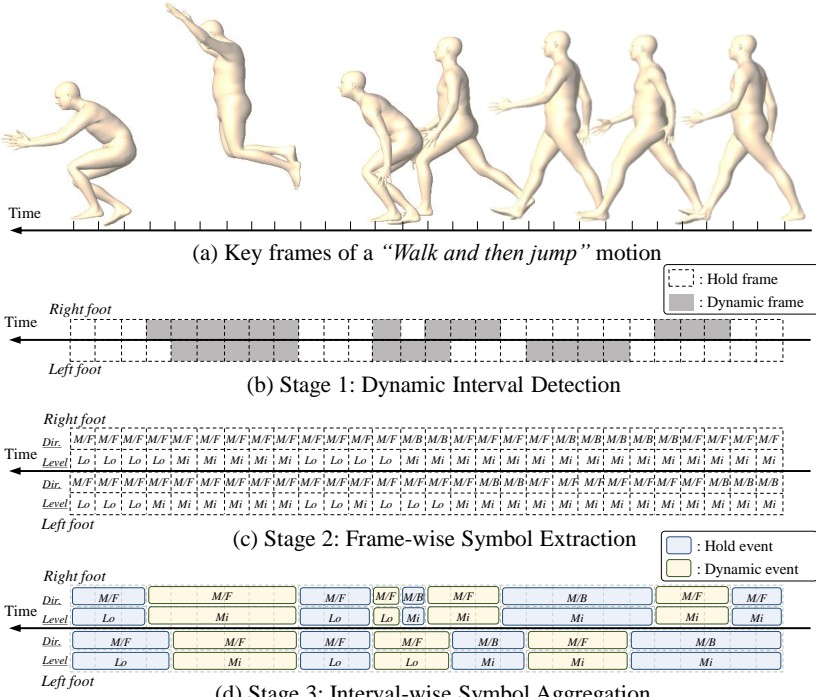

Figure 9: Illustration of extracting lower-body conceptual symbols: (a) Given a motion sequence; (b) we segment the sequence into dynamic and hold intervals by computing foot velocity; (c) in parallel, frame-wise Laban symbols are identified; (d) for each interval, aggregated frame-wise symbols yield the most representative symbol for the interval.

## A.2 AUTOMATIC LABAN CODE DETECTION WORKFLOW

As described in the main paper, the proposed workflow consists of three steps: (1) Dynamic Interval Segmentation, which identifies dynamic and hold intervals; (2) Frame-wise Symbol Extraction, which converts the pose of each frame into corresponding Laban symbols; and (3) Interval-wise Symbol Aggregation, which selects the most representative Laban symbol for each interval. As illustrated in Fig. 9, we demonstrate an example of the lower body symbol detection process. Here, we provide a detailed formulation of the Frame-wise Symbol Extraction step, including the associated predefined thresholds.

Specifically, all motion sequences are parameterised using the SMPL model (Pavlakos et al., 2019) without facial and finger key joints. Key joints such as the left/right hand, elbow, shoulder, hip, knee, foot, pelvis, and spine2 are extracted to compute the corresponding Laban symbol sequences.

### A.2.1 Details of Frame-wise Symbol Extraction

**Direction and Level symbols.** To ensure consistency, all body motions are transformed into a canonical space. This is achieved by fixing the root (pelvis) of each pose at the origin, aligning the triangular plane formed by the pelvis, right leg, and left leg key joints with the xz-plane, and orienting the body to face the negative y-axis. For each key joint (left/right hand, elbow, foot, knee), the $L_2$-norm distance to the pelvis is projected onto the x, y, and z planes, denoted as $a$, $b$, and $c$, respectively. These projected distances are compared against predefined thresholds to assign Direction and Level symbols for different body part groups (Support-L, Support-R, Upper-R, Upper-L).

For the lower body, the thresholds are as follows:

- Direction (x-axis): $a < -0.1$ is assigned 'R'; $a > 0.3$ is 'L'; otherwise, 'M'.
- Direction (y-axis): $b < -0.15$ is 'F'; $b > -0.05$ is 'B'; otherwise, 'M'.
- Level (z-axis): $0 > c > -0.8$ is 'Lo.'; $c > 0$ is 'Hi.'; otherwise, 'Mi.'.

For the upper body:

- Direction (x-axis): $a < -0.1$ is 'R'; $a > 0.3$ is 'L'; otherwise, 'M'.
- Direction (y-axis): $b < -0.2$ is 'F'; $b > 0.1$ is 'B'; otherwise, 'M'.
- Level (z-axis): $c < -0.2$ is 'Lo.'; $c > 0.1$ is 'Hi.'; otherwise, 'Mi.'.

Figure 8 illustrates the distance calculation for the left hand as an example.

**Hold symbol.** The velocity magnitude of each hand and foot is analysed. Local maxima in the x, y, and z velocity components are identified to determine the turning points of the wrist's three-dimensional trajectory. Frames in which the velocity falls below a predefined threshold are labelled as 'hold'. The threshold is set to 0.015 for the feet and 0.0005 for the hands.

**Bend symbol.** Euler angles between adjacent body segments are computed and discretised into six intervals, each spanning $30°$.

**Orientation symbol.** The facing orientation is determined by calculating the angle between the line connecting the hip key joints and the negative y-axis. The resulting angle is quantised into eight discrete directions, each spanning $45°$.

**Moving-effort symbol.** The global absolute velocity of the pelvis is computed and discretised using predefined intervals. The velocity components on the $xy$- and $yz$-planes are assigned to one of five speed categories: 0 (very slow), 1 (slow), 2 (normal), 3 (fast), and 4 (very fast), according to the following rules: For both horizontal and vertical velocity components:

- $0.1 < v \leq 0.5$: label 1;
- $0.5 < v \leq 1.0$: label 2;
- $1.0 < v \leq 2.0$: label 3;
- $v > 2.0$: label 4;
- $v \leq 0.1$: label 0.

### A.3 Details of Laban Benchmark

### A.3.1 Laban Metrics

Given the ground-truth symbol instance sequence $S$ and the generated symbol instance sequence $\hat{S}$, the proposed three metrics, including Semantic Alignment (SMT), Temporal Alignment (TMP) and Harmonious Alignment (HMN), calculate the similarity between $S$ and $\hat{S}$ using the Longest Common Subsequence (LCS) length.

Specifically, for SMT and TMP, we measure the Left/Right hand and foot body parts, while for HMN, we measure the body part pairs of [Left hand, Right hand], [Left foot, Right foot], [Left hand, Left foot], [Left hand, Right foot], [Right hand, Left foot], and [Right hand, Right foot]. In the main paper, we report the average HMN scores of [Left hand, Left foot], [Left hand, Right foot], [Right hand, Left foot], and [Right hand, Right foot] pairs due to the table scale limit.

Based on Eq. 4, a symbol instance sequence is defined by $S = \{s_t^{i,j}\}$ where $t$, $i$, $j$ denote the frame index, attribute index and Body-Part Group index, respectively. We combine the same attribute from each Body-Part Group across frames to form a duration-ignored symbol instance sequence $\tilde{S}$:

$$\tilde{S} = \{\tilde{s}_n^{i,j} \mid n \in [1, N_{i,j}], \ i \in [1, A_j] \ j \in [1, G]\}, \tag{5}$$

where $n$ denote the duration-ignored symbol instance index and $N_{i,j}$ represents the total instance number of $i$-th attribute and $j$-th Body-Part Group.

To calculate the selected body part's Laban metric, we fix the attribute index and Body-Part Group index, considering the subset $S_{i^\star,j^\star} = \{s_t^{i^\star,j^\star} \mid t \in [1, T]\}$ and the duration-ignored subset $\tilde{S}_{i^\star,j^\star} = \{\tilde{s}_n^{i^\star,j^\star} \mid n \in [1, N_{i,j}]\}$.

**Semantic Alignment (SMT)** evaluates the similarity of inter-body part symbol instances while disregarding their durations. Given an attribute index-fixed, Body-Part Group-fixed duration-ignored subset $\tilde{S}_{i^\star,j^\star}$ and its generation $\hat{\tilde{S}}_{i^\star,j^\star}$, we formulate the LCS, computing under dynamic programming as follows:

$$f(u,v) = \begin{cases} 0 & \text{if } u = 0 \text{ or } v = 0, \\ f(u-1, v-1) + 1 & \text{if } \tilde{s}_u^{i^\star,j^\star} = \hat{\tilde{s}}_v^{i^\star,j^\star}, \\ \max(f(u-1, v), f(u, v-1)) & \text{otherwise}, \end{cases} \tag{6}$$

where $u$ and $v$ denote the index of the duration-ignored symbol instance sequence, i.e., $u, v \in N_{i^\star,j^\star}$. Such that, the SMT between $\tilde{S}_{i^\star,j^\star}$ and $\hat{\tilde{S}}_{i^\star,j^\star}$ is calculated by the normalised length of the LCS:

$$\text{Sim}_{\text{SMT}}(\tilde{S}_{i^\star,j^\star}, \hat{\tilde{S}}_{i^\star,j^\star}) = \frac{f(N_{i^\star,j^\star}, \hat{N}_{i^\star,j^\star})}{\max(N_{i^\star,j^\star}, \hat{N}_{i^\star,j^\star})}. \tag{7}$$

**Temporal Alignment (TMP)** evaluates the similarity of each inter-body-part symbol instance while considering each symbol's duration, to ensure that not only the types but also the temporal extents of the motions are consistent between the ground truth and the generated sequence. To account for the duration of each symbol, the inputs are changed to symbol instance sequences $S_{i^\star,j^\star}$ and $\hat{S}_{i^\star,j^\star}$, and we modify Eq. 6 as:

$$g(u,v) = \begin{cases} 0 & \text{if } u = 0 \text{ or } v = 0, \\ g(u-1, v-1) + 1 & \text{if } s_u^{i^\star,j^\star} = \hat{s}_v^{i^\star,j^\star}, \\ \max(g(u-1, v), g(u, v-1)) & \text{otherwise}, \end{cases} \tag{8}$$

where $u$ and $v$ denote the frame index, i.e., $u, v \in T$. The Temporal Alignment score is then:

$$\text{Sim}_{\text{TMP}}(S_{i^\star,j^\star}, \hat{S}_{i^\star,j^\star}) = \frac{g(T, \hat{T})}{\max(T, \hat{T})}. \tag{9}$$

**Harmonious Alignment (HMN)** evaluates the synchronous occurrence of Laban symbols across pairs of body parts. Given a body part pair specified by $[(i_1, j_1), (i_2, j_2)]$, and their corresponding duration-ignored symbol instance sequences $\tilde{S}_{i_1,j_1}$, $\tilde{S}_{i_2,j_2}$ for the ground truth, and $\hat{\tilde{S}}_{i_1,j_1}$ and $\hat{\tilde{S}}_{i_2,j_2}$ for the generated motion, we proceed as follows. For each instance $\tilde{s}_u^{i_1,j_1} \in \tilde{S}_{i_1,j_1}$, where $1 \leq u \leq N_{i_1,j_1}$, we identify the instance $\tilde{s}_v^{i_2,j_2} \in \tilde{S}_{i_2,j_2}$, where $1 \leq v \leq N_{i_2,j_2}$, whose temporal span overlaps with $\tilde{s}_u^{i_1,j_1}$. If the intersection over union of their durations exceeds 50%, we consider $\tilde{s}_u^{i_1,j_1}$ and $\tilde{s}_v^{i_2,j_2}$ to occur synchronously. These synchronously occurring symbol pairs are collected into a sequence of combined symbol tuples, denoted by $\mathbb{S}_{j_1,j_2} = \{(\tilde{s}_u^{i_1,j_1}, \tilde{s}_v^{i_2,j_2})\}$, and similarly for the generated sequence $\hat{\mathbb{S}}_{j_1,j_2}$. Finally, the HMN similarity is computed as:

$$\text{Sim}_{\text{HMN}}(\mathbb{S}_{j_1,j_2}, \hat{\mathbb{S}}_{j_1,j_2}) = \text{Sim}_{\text{SMT}}(\mathbb{S}_{j_1,j_2}, \hat{\mathbb{S}}_{j_1,j_2}). \tag{10}$$

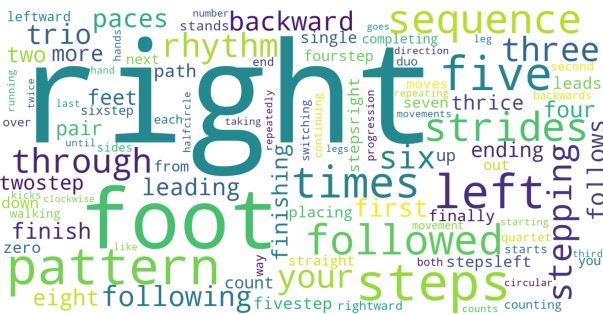

Figure 10: **WordCloud** of the most frequent words in the Locomotion descriptions, generated using the Python package `wordcloud` (Mueller, 2015).

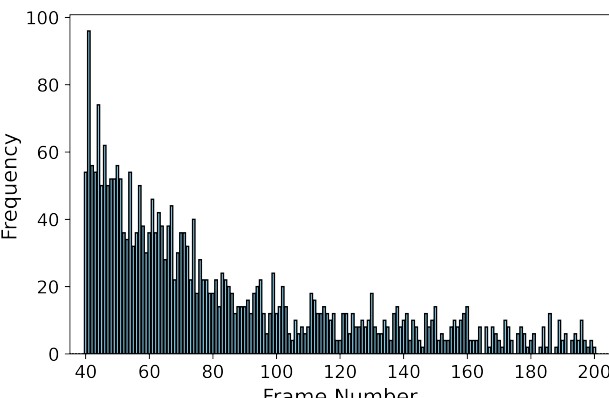

Figure 11: Distribution of the number of frames per motion sequence in the Locomotion dataset.

### A.3.2 LOCOMOTION DATASET

This section presents the construction and characteristics of our Locomotion dataset, including statistics on motion lengths, instructional text descriptions, and examples of text-motion pairs. Locomotion sequences were selected from the AMASS (Mahmood et al., 2019) dataset, covering common locomotion such as walking, running, stepping, and jumping. Following the annotation approach of BABEL-TEACH (Athanasiou et al., 2022), these actions were further decomposed into atomic actions and annotated accordingly.

**Semi-automatic Annotation.** The annotation process was conducted in a semi-automatic manner. Initially, we manually inspected rendered AMASS motion sequences to extract detailed locomotion information, such as the number of steps, their body part sequence and the action label (e.g., Walk forward: a three-step walk with the order: right, left, right). These details were stored in a JSON format with keys including "step number," "step order," and "action label." Subsequently, we utilised GPT-4.1 (Achiam et al., 2023) to generate natural language descriptions by integrating the locomotion details. The prompt used for this step is shown in the second entry of Table 6. Finally, we paired the rephrased instructional text descriptions with their corresponding motions, forming the text-motion pairs of the Locomotion dataset. In accordance with the HumanML3D (Guo et al., 2022) evaluation protocol, we restricted the motion sequence lengths to between 40 and 200 frames.

**Examples of locomotion details and text-locomotion pairs.** In the supplementary materials, we provide the complete set of instructional text descriptions that have been rephrased by a LLM. Specifically, within the "annotations" folder, each text file corresponds to a single locomotion instance and contains its associated instructional text description. For each locomotion instance, we

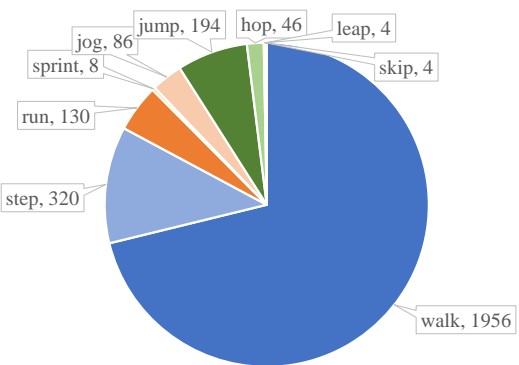

Figure 12: Distribution of action classes in the Locomotion dataset.

Table 4: The detailed structure of the Decoder in the Laban-Motion Encoder-Decoder module.

| Encoder | Num. |
|---|---|
| In Dim. | 512 |
| Feat. Dim. | 512 |
| Depth | 8 |
| Head | 8 |
| Head Dim. | 64 |
| FFN Dim. | 1024 |
| Out Dim. | 512 |

used GPT-4.1 to generate five distinct rephrasings of the description, resulting in five different annotations per text file.

**Statistics.** Following the procedure in Guo et al. (2022), all AMASS motion sequences were downsampled from 120 FPS to 20 FPS. We further filtered out locomotion sequences with fewer than 40 or more than 200 frames. As a result, the final dataset comprises 2,748 text-motion pairs, corresponding to approximately 18 hours of human motion data. Fig 12 illustrates the distribution of action labels, while Fig. 11 shows the distribution of frame lengths. Fig. 10 provides an overview of the most frequent words in the instructional text descriptions, drawn from a vocabulary of 1,220 unique words.

## A.4   EXPERIMENT DETAILS

### A.4.1   IMPLEMENTATION DETAILS

**Network Architecture.**   For the Laban-Motion Encoder-Decoder, the Encoder operates as a rule-based, non-learning process, while the Decoder is implemented as a conventional Transformer-based decoder. The Decoder incorporates standard Attention modules (Vaswani et al., 2017), including Multiheaded Self-Attention blocks (MSAs), and Feed-Forward Network blocks (FFNs), with Layernorm (LN) applied before each module. The detailed architecture of the Decoder is summarised in Table 4, where "Head" and "Head Dim." refer to the number of attention heads and the dimensionality of the features in the MSA blocks, respectively. For the Motion Generator, we adopt the configuration from Zhang et al. (2023a); Huang et al. (2024) for the Motion Generator. Specifically, a linear layer first projects the Laban code sequences, after which positional encoding is applied. The resulting sequence is then processed by a decoder-only Transformer comprising causal self-attention blocks.

**Training settings.**   The Decoder and Generator are trained with AdamW optimiser (learning rate $1 \times 10^{-4}$, batch size 512). The Decoder is trained for 200k iterations, and the Generator is trained for

100k iterations. All experiments are conducted on 1 NVIDIA L40S GPU. We select the checkpoint with the lowest FID on validation for final evaluation.

**Hyperparameters.** We adopt the motion feature extractor (Guo et al., 2022) to convert Locomotion, HumanML3D and KIT-ML motions into features of dimensions 263, 263, and 261. Locomotion and HumanML3D share the same evaluator to get text and motion embeddings. Following (Zhang et al., 2023a), we set $\lambda = 0.5$. Laban codebook contains 37 Laban categories and 158 distinct codes, with a size of $158 \times 512$. We employ two commercial LLM models named GPT-4 mini (`gpt-4.1-mini-2025-04-14`) and GPT-4 (`gpt-4.1-2025-04-14`) from Achiam et al. (2023), two open-source LLM models named Qwen3 (*qwen3-32b*) from Yang et al. (2025) and DeepSeekV3 (*deepseek-v3*) from DeepSeek-AI (2024) for composition. We use CLIP (Radford et al., 2021) (`ViT-B/32`) for text encoding.

### A.4.2 RELATED TO LARGE LANGUAGE MODELS

In the main paper, LLMs are utilised to: (1) autonomously plan and compose Laban code sequences through explicit and interpretable symbolic reasoning; and (2) semi-automatically generate instructional text descriptions for the locomotion dataset during annotation. Here, we provide a detailed explanation of the procedures for using LLMs in Laban code sequences composition.

**Prompts for high-level Laban symbolic planning.** As illustrated in Table 6 (Prompt #1), we first provide the LLM with the following information: (1) the set of Laban symbols that may be used in the composition process, (2) the expected input format from the user, and (3) the expected output format from the LLM.

The available Laban symbols are defined using a format such as: "For the support movements, the details must be selected from these 54 categories: 1: ...", where each index corresponds to a specific Laban symbol. These indices and their semantic mappings are referenced from Tables 14 and 15 for both lower and upper body movements.

The user input format is specified as a tuple, e.g., "(index, duration)", indicating the duration (in seconds) of the movement associated with a given index. The LLM is instructed to generate outputs that mimic this input format, such that each output is also represented as "(index, duration)". Finally, the indices and durations produced by the LLM are mapped back to their corresponding Laban symbols to yield Laban code sequences, which are subsequently refined temporally by the Motion Generator.

**Prompts for generating instructional text descriptions.** To construct motion–instructional text pairs, we annotate the locomotion details by manually reviewing the rendered SMPL motion videos. Specifically, we record information including the step count, step sequence, and action labels for each motion. These locomotion details are then provided to an LLM, which assists in generating instructional natural language descriptions for each motion instance. As shown in Table 6 (Prompt #2), the annotated locomotion details are translated into instructional text descriptions using the LLM. Furthermore, we include all LLM-generated instructional natural language descriptions from the locomotion dataset in the supplementary material.

### A.5 ADDITIONAL EXPERIMENTS

### A.5.1 USER STUDY

We conducted a user study to evaluate the motion generation quality of LaMoGen (configured with GPT-4.1) in comparison with two state-of-the-art models for fine-grained text-to-motion generation: ReMoDiff (Zhang et al., 2023b) and CoMo (Huang et al., 2024). Fifteen examples were randomly selected from the HumanML3D test set and rendered into video sequences. For each example, the results from LaMoGen, CoMo, and ReMoDiff were arranged side by side in a randomly determined order, forming a video triplet.

A total of 35 graduate students participated in the evaluation. Each participant was presented with all 15 video triplets and was asked to rank the three generated motions in each triplet as best, second-

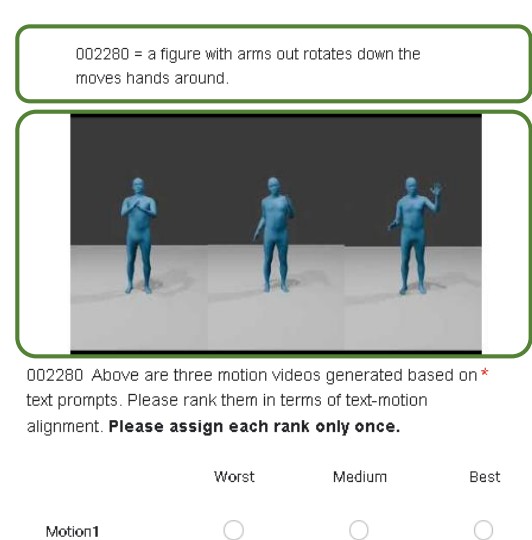

Figure 13: Illustration of the user study form.

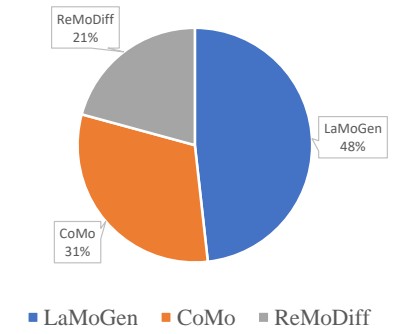

Figure 14: Illustration of the user study result.

best, and worst, based on how well they matched the provided instructions, as shown in Fig. 13. Each ranking position was assigned to exactly one method, ensuring mutual exclusivity.

**Statistical Analysis.** For quantitative analysis, we assigned scores of 3, 2, and 1 to the best, second-best, and worst rankings, respectively. The average scores for each method across all motion sequences are presented in Fig. 14. Higher scores indicate stronger user preference. As shown in Fig.14, LaMoGen achieves the highest average user preference, followed by CoMo and ReMoDiff. Nearly half of the participants favoured motions generated by LaMoGen, indicating that our method provides superior text-motion alignment.

To determine whether the observed differences in user preferences among the three methods are statistically significant, we performed a Friedman test. The results indicate a significant difference in user preference ($\chi^2 = 387.380$, $p < 0.05$). Further analysis of the mean ranks demonstrates that LaMoGen achieves the highest mean rank (mean rank = 2.80), followed by CoMo (mean rank = 1.733), and ReMoDiff (mean rank = 1.457). These results suggest that participants consistently preferred LaMoGen over the other methods, with CoMo ranked second and ReMoDiff ranked last.

Table 5: Quantitative comparisons on the HumanML3D test set, using the proposed Labanotation-based metrics: Semantic Alignment (SMT), Temporal Alignment (TMP), and Harmonious Alignment (HMN), along with Text-to-Motion metrics: R-precision Top-3 (R@3) and FID. **Bold** and underlined values indicate the best and the second-best performance, respectively.

| Method | SMT ↑ | | | | TMP ↑ | | | | HMN ↑ | | | R@3 ↑ | FID ↓ |
|---|---|---|---|---|---|---|---|---|---|---|---|---|---|
| | supL | supR | armL | armR | supL | supR | armL | armR | arm-arm | arm-sup | sup-sup | | |
| Real data | - | - | - | - | - | - | - | - | - | - | - | 0.797 | 0.002 |
| Ours (Decoder) | 0.843 | 0.843 | 0.745 | 0.731 | 0.848 | 0.848 | 0.779 | 0.765 | 0.537 | 0.631 | 0.842 | 0.793 | 0.095 |
| MDM Tevet et al. (2023) | 0.413 | 0.413 | 0.276 | 0.315 | 0.372 | 0.372 | 0.369 | 0.308 | 0.177 | 0.284 | 0.502 | 0.611 | 0.544 |
| ReMoDiff Zhang et al. (2023b) | 0.455 | 0.455 | 0.348 | 0.405 | 0.335 | 0.405 | 0.367 | 0.320 | 0.179 | 0.275 | 0.501 | 0.795 | **0.103** |
| MoDiff Zhang et al. (2024a) | 0.482 | 0.482 | 0.386 | 0.360 | 0.401 | 0.401 | 0.388 | 0.334 | 0.190 | 0.299 | 0.513 | 0.782 | 0.630 |
| CoMo Huang et al. (2024) | 0.463 | 0.463 | 0.369 | 0.401 | 0.372 | 0.372 | 0.393 | 0.346 | 0.222 | 0.321 | 0.535 | 0.790 | 0.262 |
| Ours (Bare) | 0.467 | 0.467 | 0.353 | 0.381 | 0.374 | 0.374 | 0.338 | 0.363 | 0.218 | 0.313 | 0.529 | 0.755 | 1.091 |
| Ours (GPT4.1 mini) | 0.437 | 0.437 | 0.331 | 0.354 | 0.517 | 0.517 | 0.463 | 0.445 | 0.270 | 0.371 | 0.561 | 0.779 | 0.561 |
| Ours (GPT4.1) | 0.563 | 0.563 | 0.433 | 0.456 | 0.615 | 0.615 | 0.527 | 0.544 | 0.302 | 0.411 | 0.588 | 0.796 | 0.252 |
| Ours (Laban) | **0.690** | **0.690** | **0.554** | **0.534** | **0.712** | **0.712** | **0.623** | **0.608** | **0.332** | **0.451** | **0.617** | **0.813** | 0.206 |

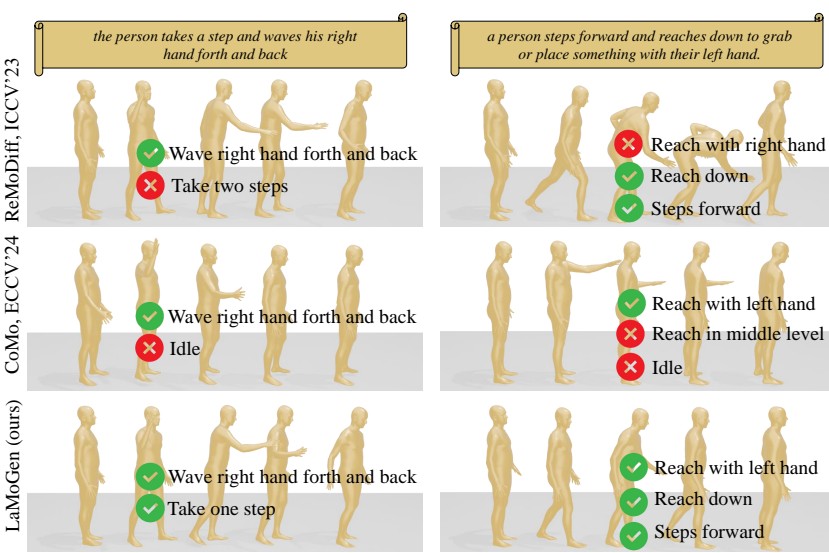

Figure 15: Qualitative comparisons on HumanML3D test sets, with motions progressing from left to right. Misalignments between text and generated motions are highlighted.

### A.5.2 QUANTITATIVE COMPARISONS

Table 5 reports the Laban metrics on the HumanML3D and KIT-ML test sets. Although the provided text descriptions are not strictly instructional—meaning they do not precisely describe the target motions and thus the calculation of Laban metrics may be of limited interpretative value—we include these results for completeness. As shown in the table, LaMoGen achieves comparable or superior performance on Laban metrics relative to other methods. This demonstrates the advantage of our approach.

### A.5.3 QUALITATIVE EXAMPLES

Additional qualitative examples are shown in Fig. 15. From this figure, we can observe that ReMoDiff and CoMo occasionally violate chronological order and struggle to specify movement timings, while our method generates text-aligned motions. Comprehensive video results, including both successful examples and failure cases, are provided in the supplementary materials. These video results were also utilised in the user study.

## A.6 LIMITATIONS AND FUTURE WORK

### A.6.1 NEED FOR A USER INTERFACE

Due to constraints in research funding and computational resources, the development of a user interface for editing Laban symbol sequences was beyond the scope of the present work. Nevertheless, we recognise that such an interface would be highly beneficial for promoting the broader adoption of the proposed LaMoGen framework and the LabanLite motion representation. As part of our future work, we plan to design and implement a user-friendly LaMoGen interface, which could be integrated as an add-on for 3D creation tools such as Blender.

### A.6.2 LEARNING COST OF LABANOTATION

Learning Labanotation does entail a certain learning cost. While incorporating a larger number of Laban symbols allows for the expression of more complex motions, it also inevitably introduces more intricate notation rules. In the main paper, we address this challenge by proposing a two-stage LLM-Guided Text-Laban-Motion Generation Module: in the first stage, either human users or LLMs compose conceptual symbol instances to represent high-level motion; in the second stage, a Kinematic Detail Augmentor further refines these conceptual instances into detailed low-level motion. This two-stage approach simplifies the notation by requiring only conceptual symbols—approximately one-fifth of the full symbol set—and thus helps to reduce the learning effort. Although a certain amount of learning is still necessary, we believe that our approach is more accessible and convenient for human editing compared to existing methods based on key frames (Liu et al., 2023; Wang et al., 2025) or key points (Wan et al., 2024).

### A.6.3 LIMITED REPRESENTATIONAL CAPACITY FOR INDIVIDUAL DIFFERENCES

As discussed in the experimental section of the main paper, the high-level abstraction provided by LabanLite may limit its ability to capture certain fine-grained semantics, such as the acceleration or deceleration of hand movements. However, if such low-level details are regarded as "personalised interpretations" of high-level semantics, LabanLite offers the advantage of disentangling motion content from style. In future work, we hope to further explore this property by treating LabanLite-based motion descriptions as representations of neutral movements, and subsequently predicting style-specific residuals for motion restyling.

### A.6.4 LIMITED REPRESENTATIONAL CAPACITY FOR HAND INTERACTION AND DANCING MOVEMENT

Currently, LabanLite defines a codebook with 158 distinct Laban codes, representing movements across the major body parts. In other words, LabanLite does not encode gestures of the fingers or toes, for now. This omission does not imply that these body parts are unimportant; finger movements are essential for human–object interaction, and toe movements are crucial for accurately recording dance motions. Presently, LabanLite is primarily designed to capture daily locomotion, but it can be readily extended to include finger and toe gestures by incorporating their corresponding Laban symbols. In fact, prior work has demonstrated that Laban symbols can facilitate fine-grained reconstruction of such motions, including finger gesture estimation(Li et al., 2024a) and dance movement reconstruction(Li et al., 2023). Therefore, as part of our future work, we plan to further extend LabanLite to facilitate applications in various domains—for example, expanding from daily movement capture to the detailed analysis of human–object interactions.

Table 6: Prompts used in our LaMoGen to process natural language via Large Language Models.

| Prompt #1: Conceptual cues composer |
| --- |
| There are five digit collections describing movements, where each line consists of: [number] [Caption] - a general description of the motion sequence. [Support] - detailed descriptions of the movements of the supporting body parts, specifically the left and right feet, using a series of triplets. [Left hand] - detailed descriptions of the movements of the left hand, using a series of tuples. [Right hand] - detailed descriptions of the movements of the right hand, using a series of tuples. In the detailed descriptions, we specify the movement details for each body part and their duration in seconds. For the support movements, the details must be selected from these 54 categories: [1: steps to rest position, ..., 54: holds in knee-flexed backward diagonally to left position]. For the hand movements, the details must be selected from these 81 categories: [1: moves close to shoulder, ..., 81: moves relatively low backward diagonally to left]. For example, for the [Support] line, the triplet list would be like: (left, 1, 0.25), (right, 2, 0.25), (left, 1, 0.25) while (right, 2, 0.25). This means that the first movement is "left foot steps to rest position in 0.25 seconds". The second movement is "right foot steps forward in 0.25 seconds". The third movement is "left foot steps to rest position in 0.25 seconds while right foot steps forward in 0.25 seconds". For the [Left hand] line, the tuple list would be like: (1, 0.5), (2, 0.2). This means that the first movement is "left hand moves close to shoulder in 0.5 seconds" and the second movement is "left hand moves forward in 0.2 seconds". For the [Right hand] line, the structure and definition are similar to [Left hand] lines. Below is the main body of the digit collection describing the movements. You should strictly imitate the following content and create only one digit collection of *YOUR_INPUT*. Reply without explanation. |
| Prompt #2: Rephraser for generating locomotion descriptions |
| Rephrase the sentence creatively *YOUR_INPUT*. the step number is *YOUR_INPUT*, with the step order: *YOUR_INPUT*. |

Table 7: Illustration of the Direction symbols and their corresponding partial semantics.

| Name | Appearance | Semantics |
| --- | --- | --- |
| Direction L/F | | Move body part left forward |
| Direction M/F | | Move body part to front |
| Direction R/F | | Move body part right forward |
| Direction L/M | | Move body part to left |
| Direction M/M | | Move body part to middle |
| Direction R/M | | Move body part to right |
| Direction L/B | | Move body part left backward |
| Direction M/B | | Move body part to back |
| Direction R/B | | Move body part right backward |

Table 8: Illustration of the Level symbols and their corresponding partial semantics.

| Name | Appearance | Semantics |
|------|------------|-----------|
| Level Hi. | | Move body part to high level |
| Level Mi. | | Move body part to mid-level |
| Level Lo. | | Move body part to low level |

Table 9: Illustration of the Hold symbols and their corresponding partial semantics. Note that if the attribute field for a symbol is left empty, it indicates that this body part is dynamic.

| Name | Appearance | Semantics |
|------|------------|-----------|
| Hold | ○ | Body part is stationary |

Table 10: Illustration of the Orientation symbols and their corresponding partial semantics. Note that according to Labanotation, each symbol does not have a specific name. In LabanLite, we simply assign them sequential names for convenience.

| Name | Appearance | Semantics |
|------|------------|-----------|
| Orient 0 | | Body part orients at around 0° |
| Orient 1 | | Body part orients at around 45° |
| Orient 2 | | Body part orients at around 90° |
| Orient 3 | | Body part orients at around 135° |
| Orient 4 | | Body part orients at around 180° |
| Orient 5 | | Body part orients at around 225° |
| Orient 6 | | Body part orients at around 270° |
| Orient 7 | | Body part orients at around 315° |

Table 11: Illustration of the Bend symbols and their corresponding partial semantics. Note that according to Labanotation, each symbol does not have a specific name. In LabanLite, we simply assign them sequential names for convenience.

| Name | Appearance | Semantics |
|------|------------|-----------|
| Bend 0 | ╳ | Body part bends at around 0° |
| Bend 1 | ╳ | Body part bends at around 30° |
| Bend 2 | ╳ | Body part bends at around 60° |
| Bend 3 | ╳ | Body part bends at around 90° |
| Bend 4 | ╳ | Body part bends at around 120° |
| Bend 5 | ╳ | Body part bends at around 150° |

Table 12: Illustration of the Moving-effort symbols and their corresponding partial semantics. Note that according to Labanotation, each symbol does not have a specific name. In LabanLite, we simply assign them sequential names for convenience. If the attribute field for a symbol is left empty, it indicates "Moving-effort 1".

| Name | Appearance | Semantics |
|------|------------|-----------|
| Moving-effort 0 | ╳ | Body part moves very slow |
| Moving-effort 1 | None | Body part moves in normal speed |
| Moving-effort 2 | ╱╱ | Body part moves fast |
| Moving-effort 3 | ╱╱ | Body part moves very fast |

Table 13: Definition of the Laban codebook. Each codebook entry is described by its index (Code Idx.), associated Body-Part Group, corresponding SMPL key joint name (SMPL Joint), relevant attribute (Attribute), a marker indicating whether the attribute is conceptual (is Concpt.), and the Laban staff column name (Staff Col.).

| Code Idx. | Body-Part Group | SMPL Joint | Attribute | is Concpt. | Staff Col. |
|---|---|---|---|---|---|
| 1 ∼ 3 | | Left foot | Direction (L/M/R) | ✓ | Left support |
| 4 ∼ 6 | | Left foot | Direction (B/M/F) | ✓ | Left support |
| 7 ∼ 9 | Support-L | Left foot | Level (Lo./Mi./Hi.) | ✓ | Left support |
| 10 ∼ 15 | | Left knee | Bend | ✗ | Left leg gesture |
| 16 ∼ 21 | | Left hip | Bend | ✗ | Left leg gesture |
| 22 ∼ 23 | | Left foot & knee & hip | Hold | ✓ | Left support |
| 23 ∼ 26 | | Right foot | Direction (L/M/R) | ✓ | Right support |
| 27 ∼ 29 | | Right foot | Direction (B/M/F) | ✓ | Right support |
| 30 ∼ 32 | Support-R | Right foot | Level (Lo./Mi./Hi.) | ✓ | Right support |
| 33 ∼ 38 | | Right knee | Bend | ✗ | Right leg gesture |
| 39 ∼ 44 | | Right hip | Bend | ✗ | Right leg gesture |
| 45 ∼ 46 | | Right foot & knee & hip | Hold | ✓ | Right support |
| 47 ∼ 54 | | Pelvis | Orient. Horiz. | ✗ | Body (Whole) |
| 55 ∼ 62 | Support-Both | Pelvis | Orient. Vert. | ✗ | Body (Whole) |
| 63 ∼ 67 | | Pelvis | Moving effort Horiz. | ✗ | Body (Whole) |
| 68 ∼ 72 | | Pelvis | Moving effort Vert. | ✗ | Body (Whole) |
| 73 ∼ 75 | | Left hand | Direction (L/M/R) | ✓ | Left hand |
| 76 ∼ 78 | | Left hand | Direction (B/M/F) | ✓ | Left hand |
| 79 ∼ 81 | Upper-L | Left hand | Level (Lo./Mi./Hi.) | ✓ | Left hand |
| 82 ∼ 87 | | Left elbow | Elbow Bend | ✗ | Left arm |
| 88 ∼ 93 | | Left shoulder | Shoulder Bend | ✗ | Left arm |
| 94 ∼ 95 | | Left hand & elbow & shoulder | Hold | ✓ | Left hand |
| 96 ∼ 98 | | Right hand | Direction (L/M/R) | ✓ | Right hand |
| 99 ∼ 101 | | Right hand | Direction (B/M/F) | ✓ | Right hand |
| 102 ∼ 104 | Upper-R | Right hand | Level (Lo./Mi./Hi.) | ✓ | Right hand |
| 105 ∼ 110 | | Right elbow | Elbow Bend | ✗ | Right arm |
| 111 ∼ 116 | | Right shoulder | Shoulder Bend | ✗ | Right arm |
| 117 ∼ 118 | | Right hand & elbow & shoulder | Hold | ✓ | Right hand |
| 119 ∼ 126 | | Head | Orient. Horiz. | ✗ | Head |
| 127 ∼ 134 | Torso | Head | Orient. Vert. | ✗ | Head |
| 135 ∼ 140 | | Spine2 | Bend | ✗ | Body (Whole) |
| 141 ∼ 143 | | Left elbow | Direction (L/M/R) | ✗ | Left arm |
| 144 ∼ 146 | Upper-L | Left elbow | Direction (B/M/F) | ✗ | Left arm |
| 147 ∼ 149 | | Left elbow | Level (Lo./Mi./Hi.) | ✗ | Left arm |
| 150 ∼ 152 | | Right elbow | Direction (L/M/R) | ✗ | Right arm |
| 153 ∼ 155 | Upper-R | Right elbow | Direction (B/M/F) | ✗ | Right arm |
| 156 ∼ 158 | | Right elbow | Level (Lo./Mi./Hi.) | ✗ | Right arm |

Table 14: Support semantic lookup table.

| Index | Semantics | Index | Semantics |
|---|---|---|---|
| 1 | steps to rest position | 28 | holds in rest position |
| 2 | steps forward | 29 | holds in forward position |
| 3 | steps backward | 30 | holds in backward position |
| 4 | steps to right | 31 | holds in right position |
| 5 | steps to left | 32 | holds in left position |
| 6 | steps forward diagonally to right | 33 | holds in forward diagonally to right position |
| 7 | steps forward diagonally to left | 34 | holds in forward diagonally to left position |
| 8 | steps backward diagonally to right | 35 | holds in backward diagonally to right position |
| 9 | steps backward diagonally to left | 36 | holds in backward diagonally to left position |
| 10 | rises | 37 | holds in the raised position |
| 11 | rises to forward | 38 | holds in the raised forward position |
| 12 | rises to backward | 39 | holds in the raised backward position |
| 13 | rises to right | 40 | holds in the raised right position |
| 14 | rises to left | 41 | holds in the raised left position |
| 15 | rises forward diagonally to right | 42 | holds in the raised forward diagonally to right position |
| 16 | rises forward diagonally to left | 43 | holds in the raised forward diagonally to left position |
| 17 | rises backward diagonally to right | 44 | holds in the raised backward diagonally to right position |
| 18 | rises backward diagonally to left | 45 | holds in the raised backward diagonally to left position |
| 19 | knee flex | 46 | holds in knee-flexed position |
| 20 | knee flex forward | 47 | holds in knee-flexed forward position |
| 21 | knee flex backward | 48 | holds in knee-flexed backward position |
| 22 | knee flex right | 49 | holds in knee-flexed right position |
| 23 | knee flex left | 50 | holds in knee-flexed left position |
| 24 | knee flex forward diagonally to right | 51 | holds in knee-flexed forward diagonally to right position |
| 25 | knee flex forward diagonally to left | 52 | holds in knee-flexed forward diagonally to left position |
| 26 | knee flex backward diagonally to right | 53 | holds in knee-flexed backward diagonally to right position |
| 27 | knee flex backward diagonally to left | 54 | holds in knee-flexed backward diagonally to left position |

Table 15: Arm semantic lookup table.

| Index | Semantics | Index | Semantics | Index | Semantics |
|---|---|---|---|---|---|
| 1 | moves close to shoulder | 28 | holds close to shoulder position | 55 | moves relatively to previous position |
| 2 | moves forward | 29 | holds forward position | 56 | moves relatively forward |
| 3 | moves backward | 30 | holds backward position | 57 | moves relatively backward |
| 4 | moves to right | 31 | holds right position | 58 | moves to relatively right |
| 5 | moves to left | 32 | holds left position | 59 | moves to relatively left |
| 6 | moves forward diagonally to right | 33 | holds forward diagonally to right position | 60 | moves relatively forward diagonally to right |
| 7 | moves forward diagonally to left | 34 | holds forward diagonally to left position | 61 | moves relatively forward diagonally to left |
| 8 | moves backward diagonally to right | 35 | holds backward diagonally to right position | 62 | moves relatively backward diagonally to right |
| 9 | moves backward diagonally to left | 36 | holds backward diagonally to left position | 63 | moves relatively backward diagonally to left |
| 10 | rises up | 37 | holds up position | 64 | moves relatively up |
| 11 | rises to up forward | 38 | holds up forward position | 65 | moves relatively up forward |
| 12 | rises to up backward | 39 | holds up backward position | 66 | moves relatively up backward |
| 13 | rises to up right | 40 | holds up right position | 67 | moves relatively up right |
| 14 | rises to up left | 41 | holds up left position | 68 | moves relatively up left |
| 15 | rises up forward diagonally to right | 42 | holds up forward diagonally to right position | 69 | moves relatively up forward diagonally to right |
| 16 | rises up forward diagonally to left | 43 | holds up forward diagonally to left position | 70 | moves relatively up forward diagonally to left |
| 17 | rises up backward diagonally to right | 44 | holds up backward diagonally to right position | 71 | moves relatively up backward diagonally to right |
| 18 | rises up backward diagonally to left | 45 | holds up backward diagonally to left position | 72 | moves relatively up backward diagonally to left |
| 19 | lowers down | 46 | holds low position | 73 | moves relatively low |
| 20 | lowers to down forward | 47 | holds low forward position | 74 | moves relatively low forward |
| 21 | lowers to down backward | 48 | holds low backward position | 75 | moves relatively low backward |
| 22 | lowers to down right | 49 | holds low right position | 76 | moves relatively low right |
| 23 | lowers to down left | 50 | holds low left position | 77 | moves relatively low left |
| 24 | lowers down forward diagonally to right | 51 | holds low forward diagonally to right position | 78 | moves relatively low forward diagonally to right |
| 25 | lowers down forward diagonally to left | 52 | holds low forward diagonally to left position | 79 | moves relatively low forward diagonally to left |
| 26 | lowers down backward diagonally to right | 53 | holds low backward diagonally to right position | 80 | moves relatively low backward diagonally to right |
| 27 | lowers down backward diagonally to left | 54 | holds low backward diagonally to left position | 81 | moves relatively low backward diagonally to left |

