# OpenReview forum: "LaMoGen: Language to Motion Generation Through LLM-Guided Symbolic Inference"
_ICLR.cc/2026/Conference — ICLR 2026 Conference Withdrawn Submission_

### Official Review · Reviewer_cmt5 · 2025-10-20

**Soundness:** 1
**Presentation:** 2
**Contribution:** 2
**Rating:** 4
**Confidence:** 4

**Summary:**

This paper introduces a two-stage text-to-motion framework built around an interpretable, Laban-inspired symbolic representation (“LabanLite”). Stage-1 retrieves top-K text–motion exemplars and prompts an LLM to produce a **conceptual** sequence of Laban symbols; Stage-2 augments that plan with **kinematic** details and decodes to full-body motion via a learnable codebook and transformer decoder. The authors also propose a Laban-annotated benchmark with three **symbolic, sequence-level** metrics intended to capture temporal and multi-part consistency. Experiments on HumanML3D / KIT-ML and the user study suggest competitive performance and improved controllability.

**Strengths:**

* **Interpretable intermediate representation:** The Laban-style codebook offers human-readable motion factors (who/what moves, where/when), aiding analysis and potential editing/conditioning.
* **Decomposed generation:** Separating **conceptual planning** from **kinematic detailing** is a clear design that addresses known issues in long-horizon consistency and controllability.
* **New evaluation perspective:** The proposed Laban metrics target temporal structure and multi-part coherence, complementing standard FID/MM-Dist/R@K.

**Weaknesses:**

* There is no quantitative or qualitative head-to-head against recent token-based/VQ approaches (TM2T, MotionGPT, MoMask, Motion-Agent). Those methods typically rely on VQ-VAE codebooks; without direct comparison, it is difficult to judge **representation power** and the efficacy–realism trade-off of the proposed Laban codebook.

* All three Laban metrics depend on the paper’s **rule-based symbol detector** and fixed thresholds. This couples the evaluation to the authors’ discretization choices. The paper does not report validation against expert Labanotation or third-party labels, leaving **detector accuracy and metric robustness** unquantified.

* The authors note that HumanML3D/KIT-ML captions are **descriptive** rather than **prescriptive**, which limits the interpretability of Laban metrics. This may leads to:

  * **Ambiguity:** A caption may correspond to multiple reasonable Laban symbolizations (different discretizations of direction/effort/shape).
  * **Evaluator shift:** Scores may reflect **motion-to-motion conceptual similarity** (detected symbols vs. ground-truth clip) more than **text-to-motion faithfulness**, since the caption doesn’t uniquely determine a Laban plan.
    Despite this caveat, these metrics are emphasized as a key advantage, which risks **over-interpreting** gains on these datasets.

* The role and value proposition of the LLM in Stage-1 are under-specified. The strongest numbers come from oracle/GT Laban plans; yet even with GT plans, the downstream pipeline is not consistently competitive against other models.

**Questions:**

1. The qualitative examples are elementary. How does the framework handle **complex/rare actions** (e.g., backflip, floor work)? Are such motions **representable** with the current Laban code set, and if not, what extensions are needed?

2. **Detector validation & design.**

   * Were the **rule thresholds** for the symbol detector empirically tuned? On what data, and by what criterion?
   * Did you conduct any evaluation on the design of the detectors?
   * How do **embedding dimension** and **number of Laban codes** affect reconstruction loss and downstream metrics?

3. Is the retrieval pool strictly train-only for each evaluation split?

---

### Official Review · Reviewer_Yn8F · 2025-10-26

**Soundness:** 2
**Presentation:** 3
**Contribution:** 2
**Rating:** 4
**Confidence:** 3

**Summary:**

LaMoGen introduces a novel framework for text-to-motion generation by using a symbolic motion language called LabanLite.
It leverages a Large Language Model (LLM) to first translate text into a high-level symbolic plan, which is then refined and decoded into a final 3D motion.
This two-stage, symbolic reasoning approach enables unprecedented control, interpretability, and accuracy for complex instructions.

**Strengths:**

1. It achieves precise control over motion details, e.g., step count and timing through a human-readable symbolic representation.
2. It uniquely uses an LLM for high-level symbolic reasoning to plan motions, separating complex logic from low-level synthesis.
3. It introduces benchmark using a set of metrics (SMT, TMP, HMN) to assess temporal, semantic, and coordination alignment.

**Weaknesses:**

1. The paper's quantitative evaluation on the widely used HumanML3D benchmark does not include comparisons with several recent and highly influential state-of-the-art models, such as MoMask and MotionGPT. Furthermore, while its performance is competitive, its FID and R-Precision scores do not consistently surpass the older baselines it was compared against.
2. The paper establishes a LabanLite codebook of size 158 but does not provide a clear rationale or empirical validation for this specific choice. An ablation study exploring how different codebook sizes affect generation quality and expressive range is crucial to demonstrate that 158 is an optimal or well-reasoned selection.
3. It needs to compare the proposed motion encoding methods with prominent learned quantization methods like Vector Quantized Variational Autoencoders (VQ-VAEs).

**Questions:**

1. Can the framework scale to generate longer, coherent motion sequences, for instance, exceeding 10 seconds or more?
2. The LabanLite system is designed for highly specific, quantitative instructions like "go five steps" or "wave three times." However, the evaluation benchmarks like HumanML3D use more general, descriptive prompts such as "a person walks in a circle." How does the model generate the corresponding motion during your quantitative experiments?

---

### Official Review · Reviewer_nPFg · 2025-10-31

**Soundness:** 3
**Presentation:** 3
**Contribution:** 3
**Rating:** 6
**Confidence:** 3

**Summary:**

This paper addresses the limitations of existing text-to-motion methods (e.g., poor temporal accuracy, weak explainability, and unreliable text-motion alignment) by introducing a symbolic reasoning paradigm. It proposes LabanLite, a human-interpretable motion representation grounded in Labanotation, which decomposes complex motions into discrete symbolic codes (conceptual symbols for high-level intent and detail symbols for execution attributes). Building on LabanLite, the authors present LaMoGen, a two-stage framework: 1) Large Language Models (LLMs) perform retrieval-augmented symbolic planning to generate conceptual motion sequences; 2) a Kinematic Detail Augmentor enriches these sequences with fine-grained attributes via autoregressive generation. The paper also introduces a Labanotation-based benchmark with three metrics (Semantic/Temporal/Harmonious Alignment) to evaluate multi-dimensional text-motion alignment. Experiments on HumanML3D, KIT-ML, and the proposed Locomotion benchmark show LaMoGen achieves state-of-the-art performance in interpretability and controllability, outperforming baselines in capturing temporal structure and compositional actions, despite slightly higher FID scores due to its high-level abstraction.

**Strengths:**

1. LabanLite’s design (rooted in Labanotation) resolves the "black-box" issue of joint text-motion embeddings. Its separation of conceptual (direction/level/hold) and detail (orientation/bend/effort) symbols enables explicit, human-readable alignment between text instructions and motion trajectories.
2. LaMoGen pioneers LLM-driven autonomous symbolic composition for motion generation, rather than using LLMs as passive decomposers. Retrieval-augmented prompting allows LLMs to reason about temporal order, action composition, and body-part coordination—addressing key weaknesses of prior methods.
3. The proposed Laban benchmark fills a critical gap in existing evaluations by measuring symbolic, temporal, and multi-body harmonious alignment. It complements standard metrics (FID, R-precision) and provides a more comprehensive assessment of text-motion consistency.
4. Experiments cover multiple datasets (in-domain and out-of-domain) and include ablation studies (LLM capability, retrieval count, masking ratio) and a user study. Results consistently demonstrate LaMoGen’s superiority in compositional generation, temporal accuracy, and user preference.

**Weaknesses:**

1. LabanLite’s high-level abstraction fails to capture individual differences (e.g., movement speed variations) and fine-grained semantics (e.g., finger/toe gestures), leading to higher FID scores compared to baselines that model low-level motion variations.
2. Performance heavily relies on LLM strength (GPT-4.1 outperforms smaller models) and is constrained by context windows—adding more than 3 retrieval examples provides no benefit and may degrade performance.
3. Labanotation requires a learning cost for human users, and the lack of a user interface limits real-world application (e.g., integration with 3D animation tools like Blender).
4.  LabanLite’s current codebook (158 codes) focuses on daily locomotion and lacks support for specialized motions (e.g., dance, fine-grained hand-object interaction), restricting its applicability to broader scenarios.

**Questions:**

1. What empirical or theoretical basis supports the selection of specific attributes (e.g., 6 bins for Bend, 5 levels for Moving-effort)? Could expanding or pruning the symbol set (e.g., adding finger/toe symbols) improve expressiveness without sacrificing computational efficiency?

2.  How does LaMoGen perform on ultra-long or ambiguous text instructions (e.g., "walk slowly for 10 steps, then jump twice while waving both hands") that exceed typical LLM context windows? Is there a scalable solution to handle such cases?

3.  The paper attributes higher FID to LabanLite’s abstraction—can a hybrid approach (e.g., combining symbolic planning with style residual learning) preserve interpretability while improving low-level motion naturalness?

---

### Note · Authors · 2025-11-13

I have read and agree with the venue's withdrawal policy on behalf of myself and my co-authors.